# Single nuclei RNA-sequencing of adult brain neurons derived from type 2 neuroblasts reveals transcriptional complexity in the insect central complex

Derek G Epiney[†], Gonzalo Morales Chaya[†], Noah R Dillon[†], Sen-Lin Lai*, Chris Q Doe*

Institute of Neuroscience, Howard Hughes Medical Institute, University of Oregon, Eugene, United States

*For correspondence:
slai@uoregon.edu (S-LL);
cdoe@uoregon.edu (CQD)

[†]These authors contributed equally to this work

Competing interest: The authors declare that no competing interests exist.

## eLife Assessment

This **important** study offers a molecular characterization of neurons and glia in the adult nervous system of the fruit fly *Drosophila melanogaster*. The study focuses on the progeny of a specific set of neural stem cells that contribute to the central complex, a conserved brain region that plays key roles in sensorimotor integration. The data are **convincing** and collected using validated methodology, generating an invaluable resource for future studies. The study will be of interest to developmental neurobiologists.

**Abstract** In both *Drosophila* and mammals, the brain contains the most diverse population of cell types of any tissue. It is generally accepted that transcriptional diversity is an early step in generating neuronal and glial diversity, followed by the establishment of a unique gene expression profile that determines morphology, connectivity, and function. In *Drosophila*, there are two types of neural stem cells, called Type 1 (T1) and Type 2 (T2) neuroblasts. The diversity of T2-derived neurons contributes a large portion of the central complex (CX), a conserved brain region that plays a role in sensorimotor integration. Recent work has revealed much of the connectome of the CX, but how this connectome is assembled remains unclear. Mapping the transcriptional diversity of T2-derived neurons is a necessary step in linking transcriptional profile to the assembly of the adult brain. Here we perform single nuclei RNA sequencing of T2 neuroblast-derived adult neurons and glia. We identify clusters containing all known classes of glia, clusters that are male/female enriched, and 161 neuron-specific clusters. We map neurotransmitter and neuropeptide expression and identify unique transcription factor combinatorial codes for each cluster. This is a necessary step that directs functional studies to determine whether each transcription factor combinatorial code specifies a distinct neuron type within the CX. We map several columnar neuron subtypes to distinct clusters and identify two neuronal classes (NPF+ and AstA+) that both map to two closely related clusters. Our data support the hypothesis that each transcriptional cluster represents one or a few closely related neuron subtypes.

## Introduction

In all organisms, the brain has arguably the most complex cellular diversity, from human (*Siletti et al., 2023*) to *Drosophila* (*Davie et al., 2018*; *Franconville et al., 2018*; *Hulse et al., 2021*). Neuronal diversity is essential for the assembly and function of the adult brain, yet the 'parts list' of different

neuronal and glial cell types remains incomplete. In *Drosophila*, the laterally positioned optic lobes have been well characterized for transcriptionally distinct neurons and glia (*Konstantinides et al., 2018*; *Konstantinides et al., 2022*; *Özel et al., 2022*), as has the central brain and ventral nerve cord (*Croset et al., 2018*; *Davie et al., 2018*; *Lago-Baldaia et al., 2023*; *Li et al., 2022*; *McLaughlin et al., 2021*; *Naidu et al., 2020*; *Nguyen et al., 2021*; *Sato and Suzuki, 2022*; *Shu et al., 2023*; *Velten et al., 2022*).

The central brain contains many diverse neurons that populate important neuropils, such as the mushroom body (MB) used for learning and memory (*Sgammeglia and Sprecher, 2022*), or the central complex (CX) used for celestial navigation and sensory-motor integration (*Fisher, 2022*), among other behaviors. The central brain neurons are all generated from neural stem cells, called neuroblasts (NBs). There are two types of NBs that generate central brain neurons: Type 1 (T1) and Type 2 (T2) NBs. T1 NBs undergo asymmetric divisions to self-renew and generate a series of ganglion mother cells (GMCs), which each produce two post-mitotic neurons (*Pollington et al., 2023*; *Yu et al., 2010*; *Yu et al., 2013*). There are ~100 T1 NBs per larval central brain lobe that each generate 20–100 neurons and glia (*Ito et al., 2013*; *Yu et al., 2013*). In addition, there are Type 0 (T0) NBs which generate post-mitotic neuron progeny (*Baumgardt et al., 2014*); T0 NB lineages yet to be documented in the central brain. T2 NBs have a more complex division pattern than T1 NBs (*Bello et al., 2008*; *Boone and Doe, 2008*; *Bowman et al., 2008*). T2 NBs undergo asymmetric division to self-renew and generate an intermediate neural progenitor (INP); each INP undergoes 4–6 divisions to self-renew and generate a GMC and its two neuron or glial progeny (*Bello et al., 2008*; *Boone and Doe, 2008*; *Bowman et al., 2008*). Thus, each T2 NB division will generate 8–12 progeny cells. Most T2 NBs generate ~500 neurons (*Yu et al., 2013*), have highly complex neuronal cell types based on morphology and connectivity (*Hulse et al., 2020*; *Wang et al., 2014*; *Yang et al., 2013*; *Yu et al., 2013*), and produce the columnar neurons of the adult CX (*Boyan and Williams, 2011*; *Kandimalla et al., 2023*). Subsequently, we will call neurons born from type I NBs 'T1-derived' and neurons born from type 2 NBs 'T2-derived'.

Although single-cell RNA-seq (scRNA-seq) has been done on adult central brain neurons and glia (*Croset et al., 2018*), the transcriptomic profile of T1 versus T2 neuronal and glial progeny has not yet been characterized. Furthermore, the T2 lineages generate diverse neuronal progeny (*Hulse et al., 2020*; *Wang et al., 2014*; *Yang et al., 2013*; *Yu et al., 2013*) but focused transcriptional profiling of T2-derived progeny has not been reported. In this report, we generate a single nuclei (snRNA-seq) atlas of T2-derived progeny of the adult central brain. We use several complementary methods to link neuron identity to transcriptional clusters. Our data will facilitate linking neuronal transcriptomes with connectomes to gain a molecular understanding of CX development, connectivity, and behavior.

## Results

### Generation of a transcriptomic atlas of adult central brain neurons and glia

The adult central brain (i.e. brain without optic lobes) is composed of neuronal and glial progeny derived from both T1 and T2 NBs. We first generated a transcriptomic atlas of all central brain neurons derived from both T1 and T2 NBs. The flies were of a genotype that gave permanent lineage tracing of T2 NB-derived progeny in the adult (*Figure 1A and A'*). This allowed us to first analyze all central brain T1- and T2-derived neurons and glia (*Figure 1*) and then focus on neurons and glia produced specifically by T2-derived lineages.

We dissociated nuclei from 7-day-old adult fly central brains with optic lobes removed using methods previously described (*Li et al., 2022*; *McLaughlin et al., 2021*). The cDNA libraries were then prepared from dissociated nuclei with split-pool-based barcoding for single-nuclei transcription profiling (*Rosenberg et al., 2018*). We sequenced a total of 30,699 nuclei at 589 median genes per nuclei from the T1+T2 central brain.

To distinguish T2-derived progeny from T1-derived progeny, we expressed *worniu-Gal4,asense-Gal80* to drive expression of FLP recombinase (FLP) specifically in T2 NBs. This resulted in the flip out of a stop cassette and thus continuous expression of Gal4 under the ubiquitous *actin5C* enhancer. After removal of the stop cassette in T2 NBs, the *actin5c-Gal4* can continuously drive expression of the reporter genes (RFP or GFP) and FLP in the T2-derived neuronal and glial progeny in adult brains

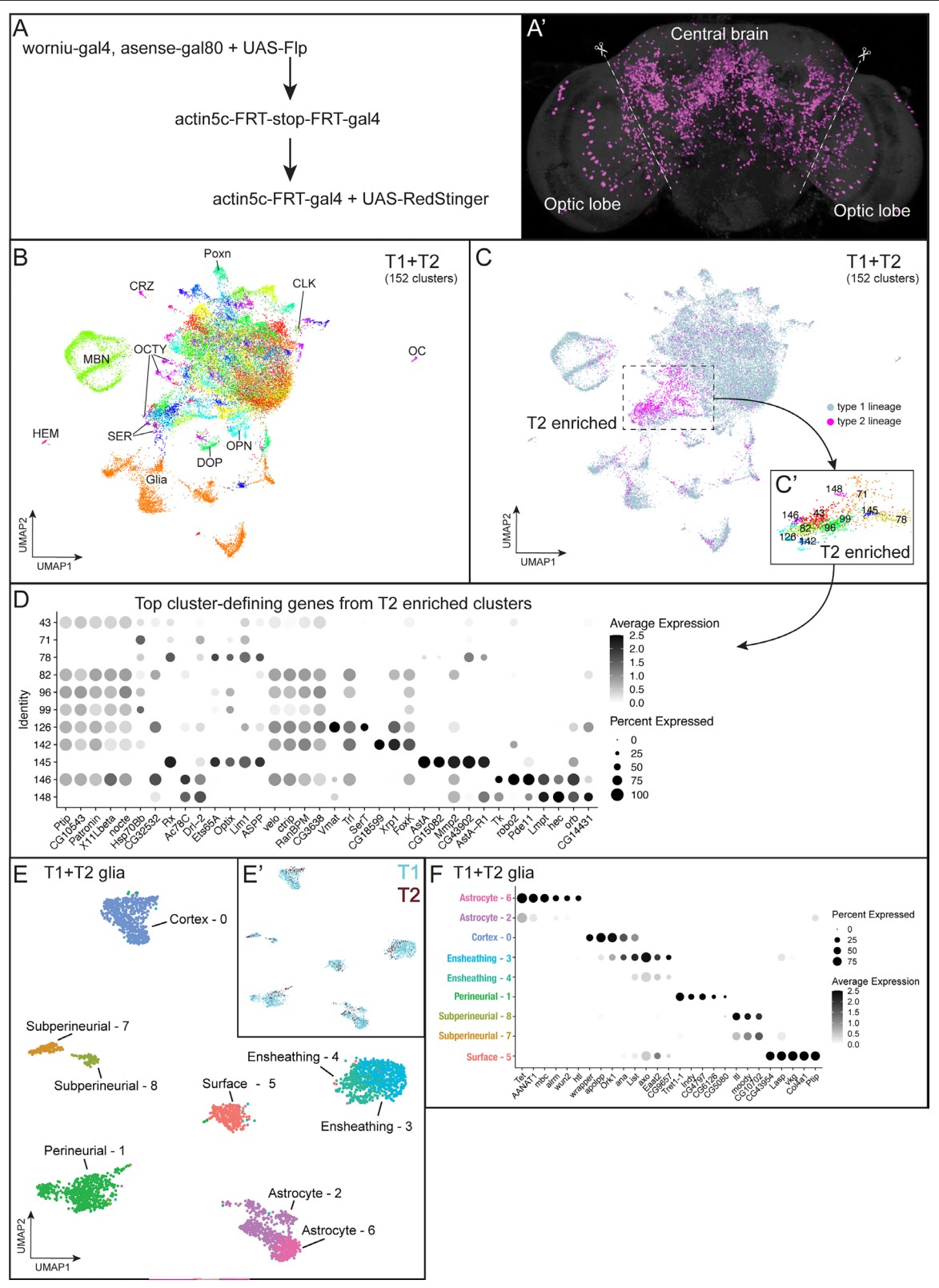

**Figure 1.** Cell atlas of central brain with single nuclei RNAseq. (**A-A'**) Genetics (**A**) to label progeny (**A'**) derived from T2 NBs. Dashed lines show the boundary of optic lobes and central brain, and the optic lobes were removed during dissection. (**B**) Central brain atlas labeled with known cell types. Abbreviations: CLK, clock neurons; CRZ, Corazonergic neurons; DOP, dopaminergic neurons; HEM, hemocytes; MBN, mushroom body neurons; OCTY, octopaminergic-tyraminergic neurons; OC, ocelli; OL, optic lobe; OPN, olfactory projection neurons; SER: serotoninergic neurons. (**C**) Central brain atlas

*Figure 1 continued on next page*

*Figure 1 continued*

labeled by NB (T1 or T2) lineage. Dash line-outlined box shows the region enriched with the cells derived from T2 NBs, and the identity are shown at the bottom-right box. (**D**) Dot plot of top 5 marker genes of the T2-enriched clusters. (**E**) Atlas of central brain glia labeled with known cell types. (**E′**) T1 and T2 derived cells colored cyan and red respect␒ly. (**F**) Dot plot of known and top marker genes of glial clusters.

The online version of this article includes the following figure supplement(s) for figure 1:

**Figure supplement 1.** Nuclei numbers T1 vs T2.

(*Figure 1A*). We assigned 3,125 nuclei which have expression of RFP, GFP, or FLP as being derived from T2 lineages. We assigned 27,574 triple-negative nuclei as derived from the T1 lineages. We used Seurat for filtering, integrating, and clustering the atlas to provide transcriptionally unique cell clusters (*Stuart et al., 2019*). We used uniform manifold approximation and projection (UMAP)-based dimension reduction to visualize the 152 clusters (*Figure 1B*). The most enriched or least expressed genes (marker genes) for each cluster are shown in *Supplementary file 1*.

We used known markers (*Supplementary file 2*; *Croset et al., 2018*; *Davie et al., 2018*) to identify distinct cell types in the central brain, including glia, mushroom body neurons, olfactory projection neurons, clock neurons, Poxn+ neurons, serotonergic neurons, dopaminergic neurons, octopaminergic neurons, corazonergic neurons, hemocytes, and ocelli (*Figure 1B*; *Supplementary file 1*). We did not observe any cluster exclusively containing progeny from T1 or T2 NB lineages (*Figure 1C*, *Figure 1—figure supplement 1*). We conclude that both T1 and T2 generate transcriptionally similar cells, despite their different developmental origins. Alternatively, deeper sequencing may resolve a single cluster into two clusters with each derived from T1 or T2 lineages. We next searched for cluster-defining genes in the T2 enriched clusters (*Figure 1C′*) and found high expression of *AstA* (*Allatostatin A*), *SerT* (*Serotonin transporter*), *Tk* (*Tachykinin*), and *Vmat* (*Vesicular monoamine transporter*) (*Figure 1D*). These enriched marker genes suggest that *AstA*+ neurons, serotonergic neurons, and *TK*+ neurons may be primarily derived from T2 lineages.

We explored the central brain atlas for glial cell types and their gene expression. We focused on the 3409 glial nuclei from clusters that expressed the pan-glial marker *repo* (*Campbell et al., 1994*; *Xiong et al., 1994*) in the T1+T2 atlas (*Figure 1E*). Each cluster contained a mix of T1 and T2 glial progeny (*Figure 1E′*). We identified six known glial cell types (astrocytes, cortex, ensheathing, surface glial and the two subtypes: perineurial and subperineurial) based on canonical glial makers (*Figure 1F*; *Supplementary file 3*). Similar to a previous glial cell atlas (*Lago-Baldaia et al., 2023*), we found some glial subtypes (astrocytes, ensheathing, and subperineurial) mapping to multiple clusters (*Figure 1E and F*).We identified a glial cluster that expressed genes associated with extracellular matrix, *vkg* and *Col4a1* (*Figure 1F*), which have previously been identified as pan surface glial markers (*DeSalvo et al., 2014*; *Hindle and Bainton, 2014*). Interestingly, the two surface glial subtypes, perineurial and subperineurial clusters, do not express these markers at cluster defining levels, and conversely the surface glia cluster 5 does not express perineurial or subperineurial specific makers (*Figure 1F*, *Supplementary file 4*). Differential gene expression analysis for all genes between T1 and T2 glial progeny did not show differences across any glial cell types or clusters (*Supplementary file 5*). We conclude that the adult central brain contains known glial cell types with no differences in gene expression between T1 and T2-derived glia.

## Generation of a T1- and T2-specific cell atlas

We explored the diversity of T1- and T2-derived neurons by generating T1- and T2-specific cell atlases. We identified T1-derived neurons by bioinformatically *excluding* cells co-expressing T2-specific markers *FLP+/GFP+/RFP+* plus *repo+* glial clusters. We then generated a T1 neuron atlas containing 22,807 T1-derived neurons that form 114 clusters. Marker genes of each cluster are shown in *Supplementary file 5*. We identified the neurons that are known to be generated by T1 NBs, including MB neurons and olfactory projection neurons (*Croset et al., 2018*; *Davie et al., 2018*; *Li et al., 2022*; *Supplementary file 2*). We identified other T1-derived neurons, including clock neurons, Poxn+ neurons, and neurons that release dopamine, serotonin, octopamine/tyramine, and other neuropeptides (*Figure 2—figure supplement 1*, *Supplementary file 6*; *Croset et al., 2018*; *Davie et al., 2018*). We gathered a list of genes that represented the 10 most enriched genes from each cluster and calculated the scaled averaged expression of each gene from each cluster (*Figure 2—figure supplement 1*). Each unique combination of enriched genes could be referred to as cluster markers.

We conclude that our T1-derived nuclei contain the expected abundance of neuronal diversity seen in previous scRNA-seq atlases (*Croset et al., 2018*; *Davie et al., 2018*; *Li et al., 2022*).

In our whole brain atlas described above, T2-derived cells represent a minor contribution to the atlas due to their low nuclei numbers relative to the T1-derived neurons (*Raji and Potter, 2021*). To generate a comprehensive T2 atlas, we needed to increase the percentage of T2-derived nuclei for RNAseq. We used fluorescence-activated cell sorting (FACS) to select nuclei that are labeled by T2-specific permanent lineage tracing (*Figure 1A and B*). We then selected the nuclei in silico that showed expression of UAS-transgenes (FLP, GFP, or RFP, see above) to eliminate contamination of T1 nuclei during FACS. In total, we sequenced 61,118 T2 nuclei. We included 3125 UAS transgene-labeled nuclei from the dissociated central brain without sorting from the T1+T2 atlas (see above). We used integration in Seurat to generate an atlas containing 64,243 nuclei that formed 198 clusters. T2 NBs have been estimated to produced ~5000 neurons/glia in the central brain (*Ito et al., 2013*; *Yang et al., 2013*), with ~1800 neurons in the central complex (*Hulse et al., 2020*; *Schlegel et al., 2024*), giving our atlas ~12 x coverage. We next filtered out the *repo*+ glial cell clusters to generate a T2 neuron cell atlas with 50,148 non-glial nuclei forming 161 clusters (*Figure 2A*). Marker genes of each cluster are shown in *Supplementary file 8*; *Marsh, 2024*.

We calculated the scaled average expression of the top 10 most enriched genes from each T2 neuron cluster (*Supplementary file 8*). Each set of genes could serve as cluster marker genes (*Figure 2B*). We conclude that each cluster within our T2 neuron atlas represents a transcriptionally unique cell type. For the remainder of the paper, we focus solely on the diversity of T2-derived neurons and glia.

## T2 neuroblasts generate all major classes of fast-acting neurotransmitters

An important aspect of neuronal identity and function is fast-acting neurotransmitter expression. We determined the neuronal populations that expressed seven fast-acting neurotransmitters: glutamatergic, cholinergic, GABAergic, tyraminergic, dopaminergic, serotonergic, and octopaminergic in T2-derived neurons (*Figure 3A–G*). In neurons, we found that cholinergic neurons were most abundant (21%), followed by glutamatergic neurons (12%), GABAergic neurons (9%), tyraminergic neurons (2.6%), dopaminergic neurons (1.2%), serotonergic neurons (1.2%), and octopaminergic neurons (0.7%). Additionally, 12% of neurons were co-expressing two or three neurotransmitters. The remaining 40% in our atlas were not expressing any neurotransmitter with a log normalized expression <2 (*Figure 3H*); this is similar to the ratios in the adult midbrain (*Croset et al., 2018*) and optic lobes (*Konstantinides et al., 2022*). The nuclei co-expressing two or more fast-acting neurotransmitters may reflect an authentic feature of these neurons, as see in other systems (see Discussion). Alternatively, our double- and triple-positive neurons may be false positives (see Discussion).

## T2 neuroblasts generate all major classes of glia

We next explored the complete T2 atlas for glial cell types and their gene expression. We subclustered from the T2 atlas for 12,315 glial nuclei from clusters that expressed the pan glial marker *repo* (*Figure 4A*; *Campbell et al., 1994*; *Xiong et al., 1994*). Similar to our T1+T2 glial atlas, we identified glial cell types: astrocytes, cortex, ensheathing, astrocyte-like, and two surface glial subtypes, perineurial and subperineurial, based on canonical glial makers (*Figure 4B*; *Supplementary file 9*). In line with our T1+T2 atlas and previous glia cell atlas (*Lago-Baldaia et al., 2023*), some subtypes mapped to several subclusters including ensheathing, astrocytes, and chiasm (*Figure 4A–B*). Interestingly, we identified chiasm glia that may represent the migratory chiasm glia generated from the DL1 lineage (*Viktorin et al., 2013*; see Discussion). We conclude that our T2 atlas contains the expected glial cell types.

## T2 neuroblasts generate sex-biased cell types

When generating our atlas, we dissected both female and male adult brains that were processed as separate samples in order to differentiate the sex of nuclei in the snRNAseq atlas. We noticed that several cell types in the T2 glial atlas showed unequal number of male and female nuclei (*Figure 5A–B and E–F*). We identified sex-biased clusters by normalizing the number of input nuclei between male and female samples and compared the proportion of nuclei by sex within each cluster to identify

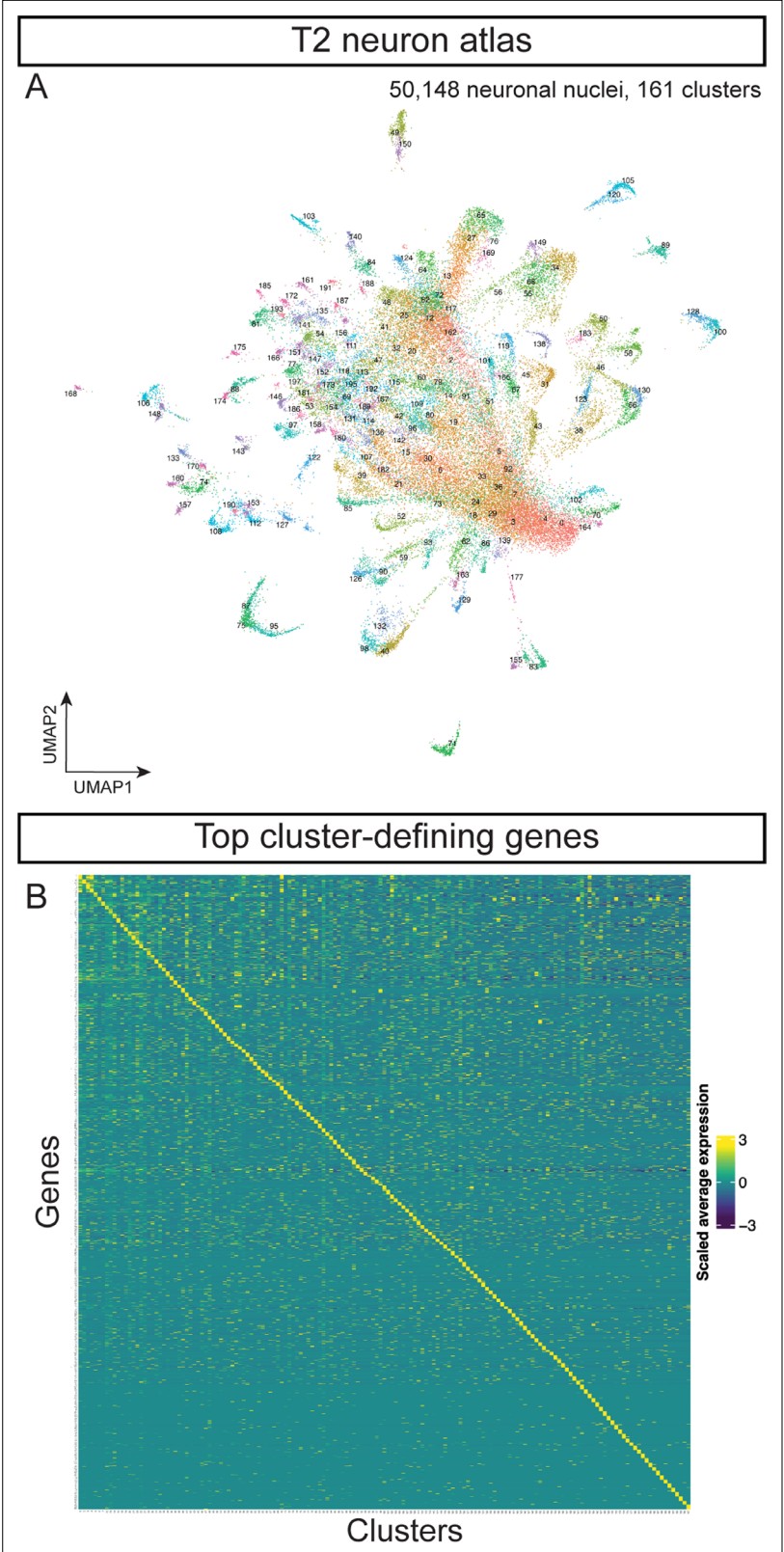

**Figure 2.** Cell atlas of neurons produced by T1 or T2 NBs. (**A**) Cell atlas from T2 NBs. (**B**) Heatmap of scaled average expression of top 10 markers genes from each T2 neuronal cluster.

The online version of this article includes the following figure supplement(s) for figure 2:

**Figure supplement 1.** T1 neuron atlas.

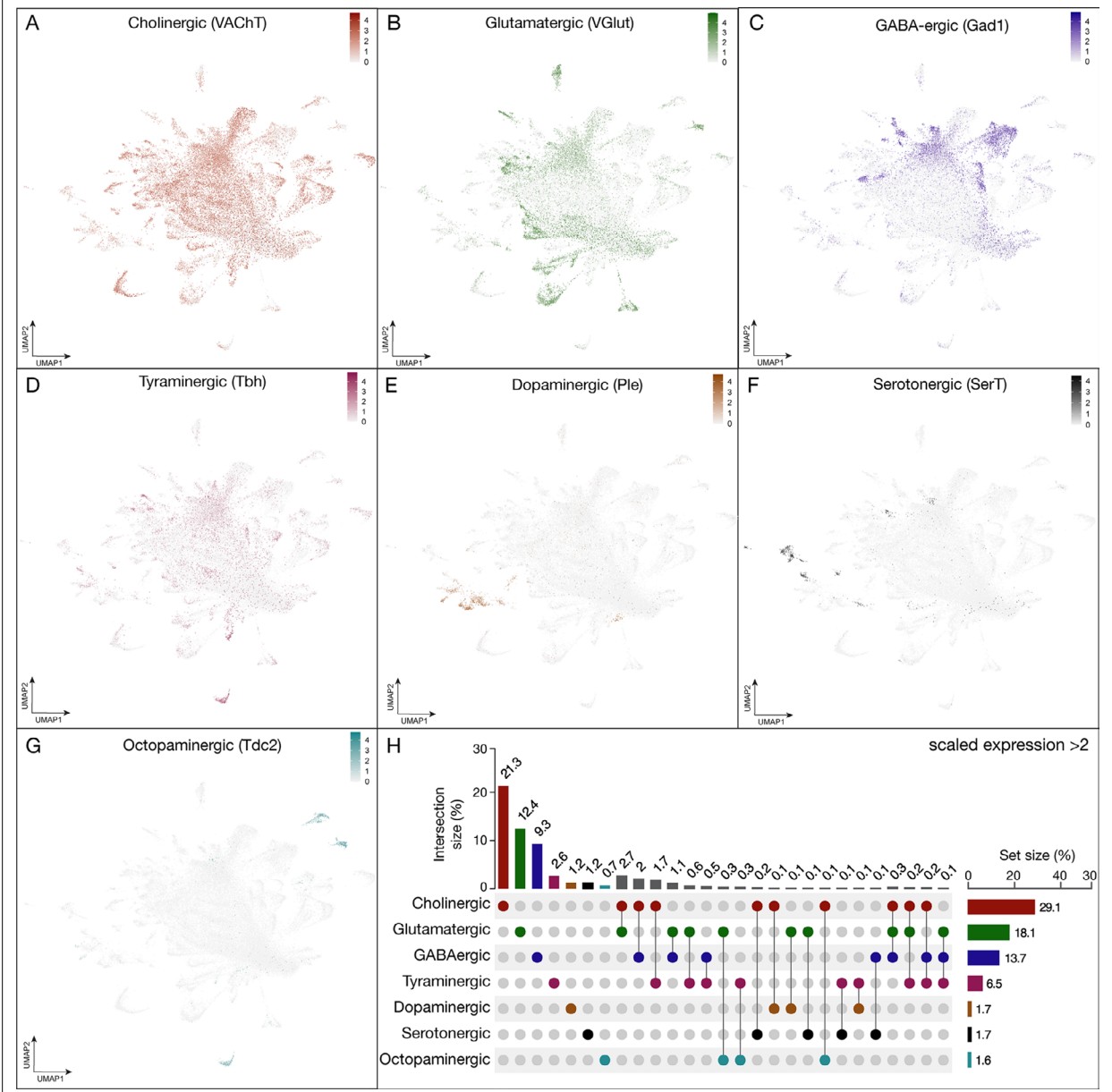

**Figure 3.** Expression of fast-acting neurotransmitters in neurons derived from T2NSC lineages. (**A–G**) UMAP distribution plots demonstrate the expression of the following neurotransmitters: (**A**) *vesicular glutamate transporter* (VGlut, glutamatergic neurons), (**B**) *vesicular acetylcholine transporter* (VAChT, cholinergic neurons), (**C**) *glutamic acid decarboxylase 1* (Gad1, GABAergic neurons), (**D**) *tyramine β hydroxylase* (Tbh, tyraminergic neurons), (**E**) *tyrosine 3-monooxygenase* (Ple, dopaminergic neurons), (**F**) *serotonin transporter* (SerT, serotonergic neurons) (**G**) *tyrosine decarboxylase* (Tdc2, octopaminergic neurons). All plots have a minimum cutoff value set at 0. (**H**) The UpSet quantifies the number of cells in the atlas that express each neurotransmitter gene with a scaled expression >2 (*Gu, 2022*).

sex-biased clusters (*Figure 5B and F*). We found that the T2 glial atlas contained two female enriched clusters: ensheathing/astrocyte (cluster 7) and chiasm (cluster 5), and one male enriched cluster: the astrocyte-like (cluster 3; *Figure 5B*). We determined differential genes expressed between male and female nuclei across all glia (*Figure 5C*; *Supplementary file 10*). We found female nuclei expressed higher levels of genes including the female-specific genes *yp1, yp2,* and *yp3* (*Figure 5C*; *Warren et al., 1979*). Additionally, female nuclei were enriched for *dsx* (*Supplementary file 10*).Male glial nuclei expressed higher levels of genes including the male-specific genes *lncRNA:rox1/2* and *fru* (*Figure 5C*; *Supplementary file 10*; *Amrein and Axel, 1997*; *Meller et al., 1997*; *Ryner et al., 1996*).We next looked at expression within the sex-biased clusters. We found similar results that male nuclei expressed

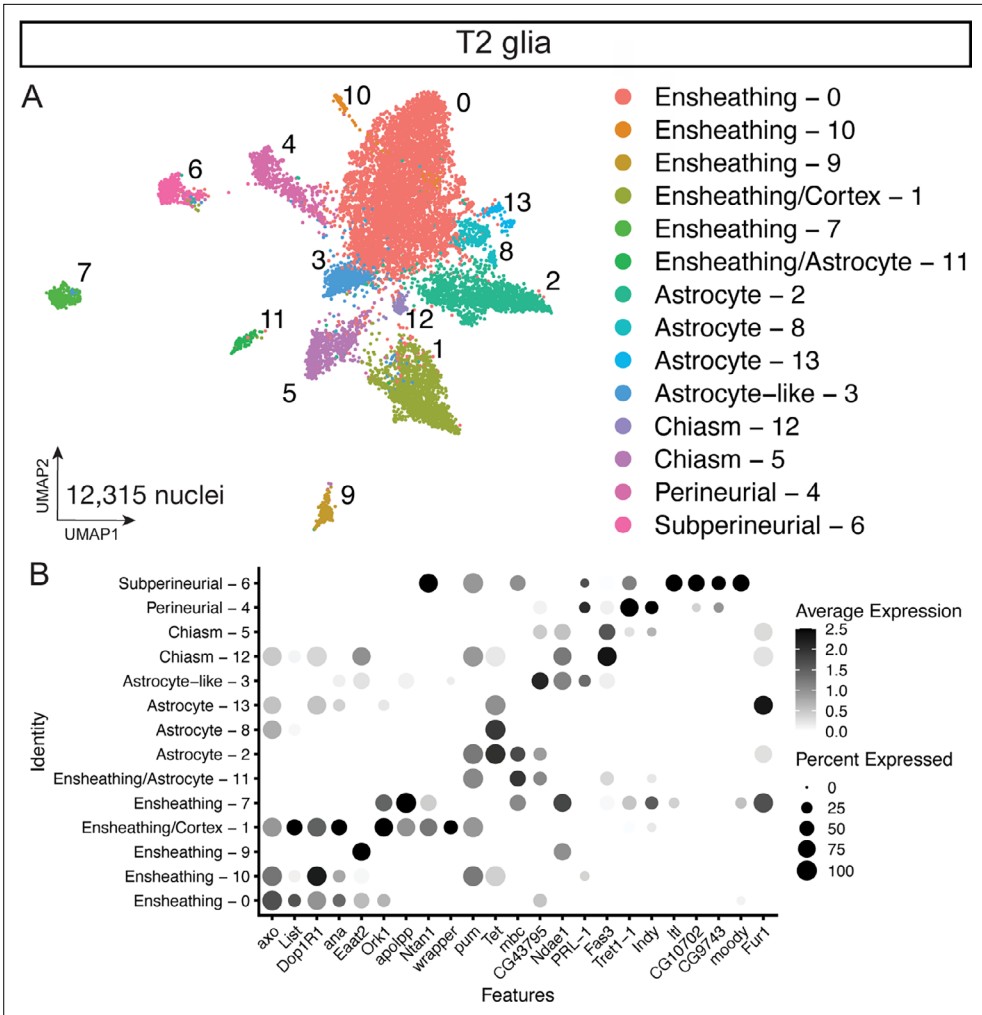

**Figure 4.** Glial cell types present in the T2-derived cell atlas. (**A**) Sub-clustered of 12,315 nuclei from Repo + T2 clusters in UMAP distribution. (**B**) Dot plot of validated glial subtype markers to identify glial clusters by differential gene expression.

high levels of the male-specific *lncRNA:roX1/2* and low expression of the female-specific genes *yp1, yp2,* and *yp3* compared to the female nuclei (*Figure 5D*; see Discussion). We conclude that male and female adult T2 glia have sex-specific differences in gene expression within the same glial cell type.

We next explored sex differences among neurons (*Figure 5E–E''*). We identified 14 neuronal clusters with disproportionate amounts of male and female nuclei (*Figure 5F*). We first tested the differential genes expressed between male and female nuclei across all T2-derived neurons (*Figure 5G*; *Supplementary file 11*). Similar to the T2 glia sex differences, we found female nuclei expressed higher levels of the female-specific genes *yp1, yp2,* and *yp3* (*Figure 5G*; *Warren et al., 1979*). Additionally, female nuclei were enriched for *dsx* (*Supplementary file 11*). Male neuronal nuclei expressed higher levels of the male-specific genes *lncRNA:rox1/2* and *fru* (*Figure 5G*; *Supplementary file 11*; *Amrein and Axel, 1997*; *Meller et al., 1997*; *Ryner et al., 1996*). We next looked at expression levels within the sex-biased clusters. We found similar results that male nuclei expressed high levels of the male-specific *lncRNA:roX1/2* and low expression of the female-specific genes *yp1, yp2,* and *yp3* compared to the female nuclei (*Figure 5H*; see Discussion). We conclude that male and female adult T2 neurons have sex-specific differences in gene expression within the same neuronal subtype.

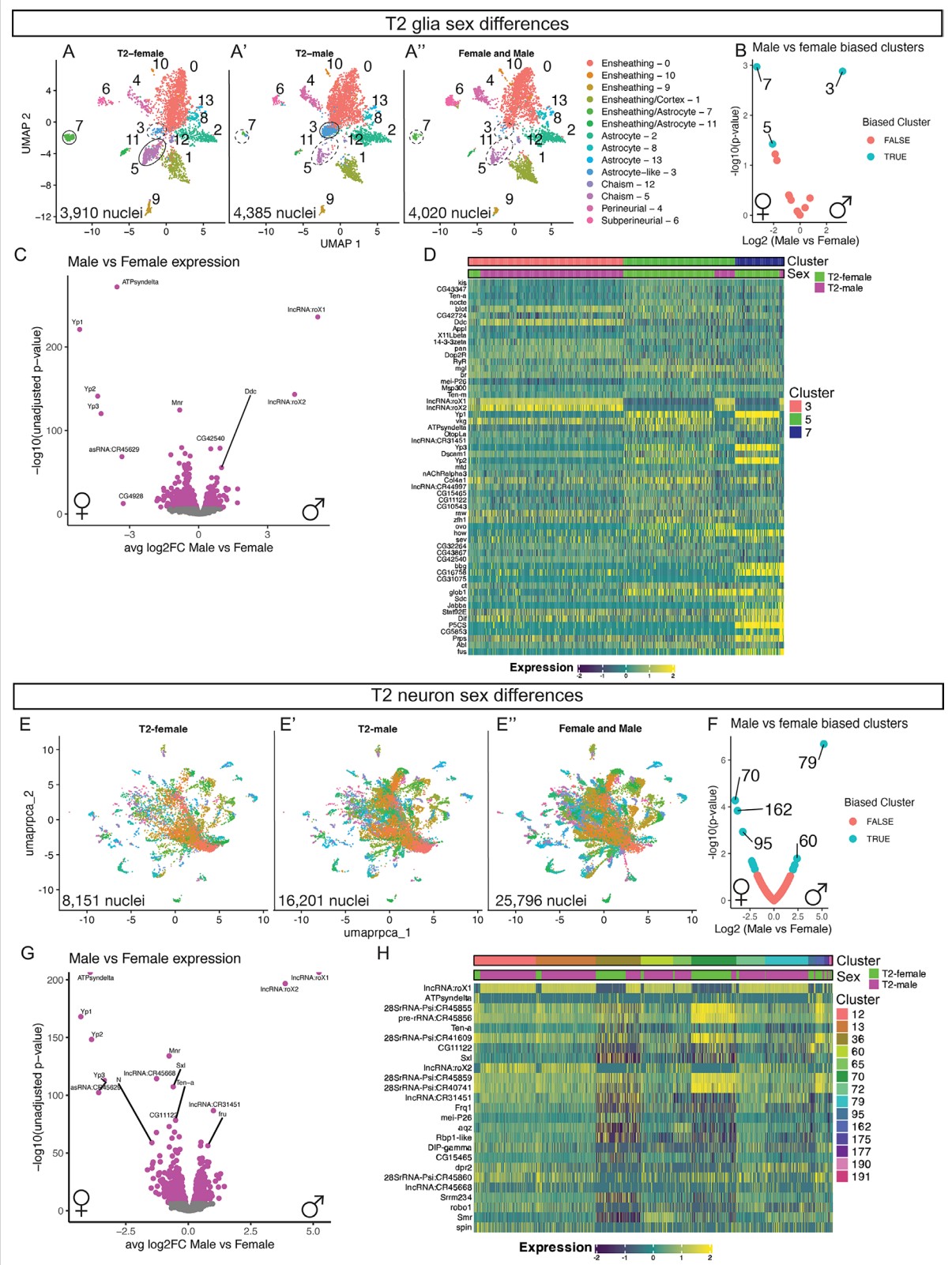

**Figure 5.** Sex differences present in the T2-derived glia and neurons. (**A-A''**) T2 glia in UMAP distribution across samples: (**A**) T2 Female (3,910 nuclei), (**A'**) T2 Male (4,385 nuclei), (**A''**) Female and Male mixed (4,020 nuclei). (**B**) Biased clusters for male to female ratio for number of nuclei in the T2 glia clusters. (**C**) Differential expression between male and female T2 glia. (**D**) Heatmap of top differential gene expression between male and female nuclei within glial clusters. (**E-E''**) T2 neuron in UMAP distribution across samples: (**E**) T2 Female (8,151 nuclei), (**E'**) T2 Male (16,201 nuclei), (**E''**) Female and

*Figure 5 continued on next page*

Male mixed (25,796 nuclei). (**F**) Biased clusters for male to female ratio for number of nuclei in the T2 neuron clusters. (**G**) Differential expression between male and female T2 neurons. (**H**) Heatmap of top differential gene expression between male and female nuclei within neuron clusters.

## T2 neuroblasts generate neurons expressing a diverse array of neuropeptides

Neuropeptides are often used as markers to distinguish different neuronal populations. We examined whether individual neuropeptide-encoding genes were exclusively expressed within single clusters, representing single neuronal subtypes. Among the 49 neuropeptides in *Drosophila*, we identified 13 with enriched expression in a limited number of clusters, making them suitable as cluster-defining markers (*Figure 6A*). For all clusters significantly expressing cluster-defining neuropeptides, we assessed their co-expression with specific fast-acting neurotransmitters (*Figure 6A*). We found that all neuropeptide-expressing clusters, co-express one or more neurotransmitters (see *Figure 3H*).

These data raise several questions. What is the relationship between neuropeptide expression and cluster identity? Can two distinct clusters express the same neuropeptide? Can two neuropeptides be co-expressed by the same cluster? Our data show that all of these patterns exist. For example, in some cases, one cluster expresses two or more neuropeptide genes (e.g. clusters 100 and 128 express

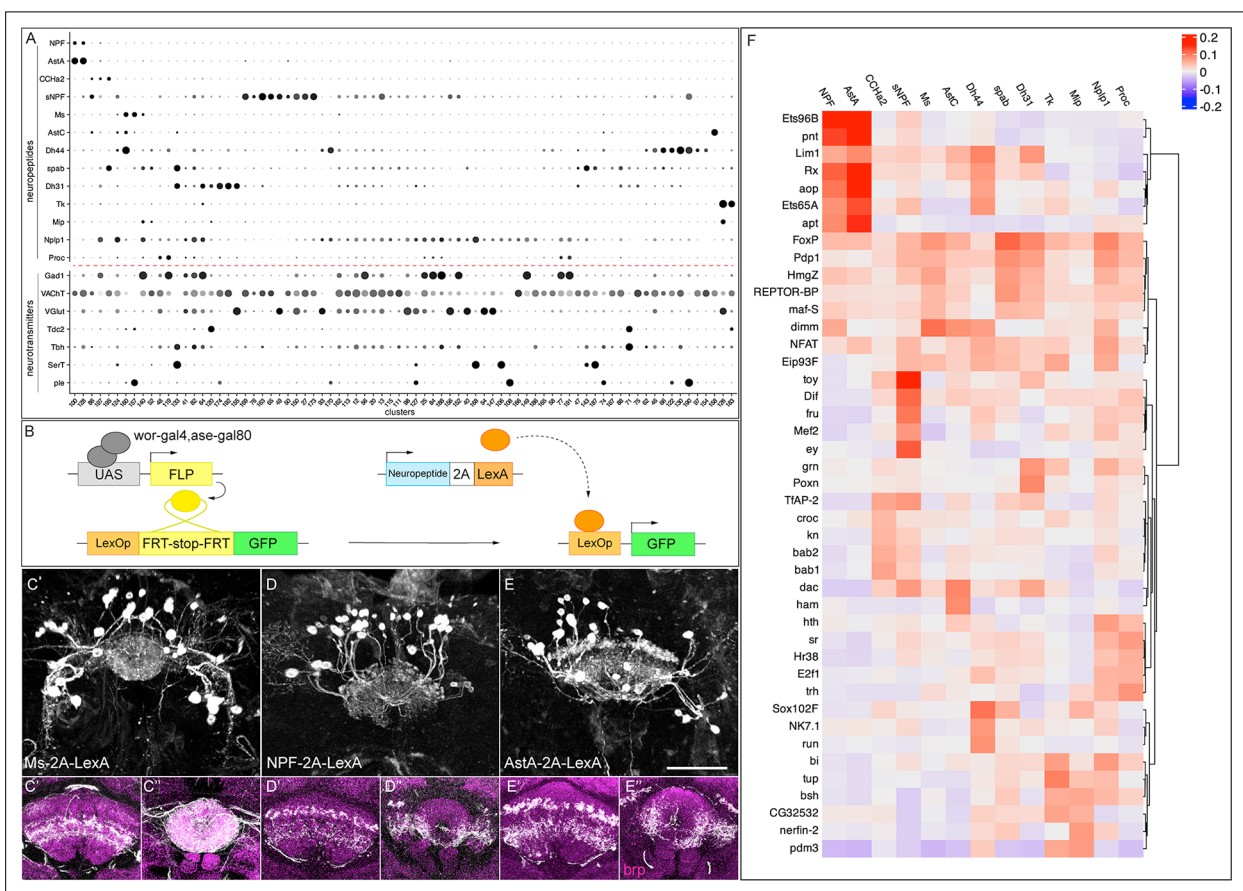

**Figure 6.** Neuropeptide expression in T2-derived neurons. (**A**) Dot plot showing expression of 13 neuropeptides and their co-expression with the 7 fast-acting neurotransmitters (red dashed line) across selected clusters. (**B**) Genetic scheme to map neuropeptide expression in T2-derived adult neurons. (**C–E**) Neuropeptide-expressing neurons labeled with GFP in three-dimensional projections. (**C'–E'**) Fan-shaped body projections. (**C''-E''**) Ellipsoid body projections. nc82 counterstains (magenta) in the brain for neuropil projections. (**F**) Heatmap showing the top 5 transcription factors most strongly correlated with each neuropeptide across all cells (*Sigorelli, 2024*). Scale bar represents 20 μm.

The online version of this article includes the following figure supplement(s) for figure 6:

**Figure supplement 1.** Expression correlation coefficient of transcription factors and neuropeptides.

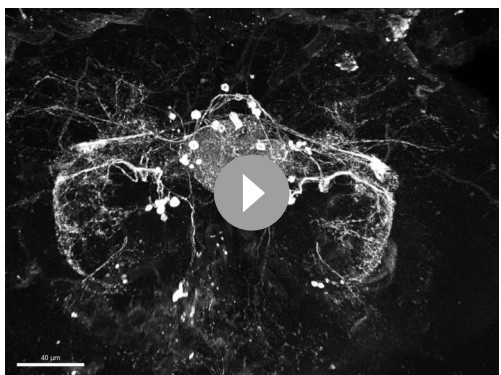

**Video 1.** Rotation of Imaris three-dimensional view of Ms-2A-LexA expression. Genotype: 20xUAS-flp;worniu-gal4,asense-gal80; lexAop-FRT-stop-FRT-myr::gfp x Ms-2A-LexA.

https://elifesciences.org/articles/105896/figures#video1

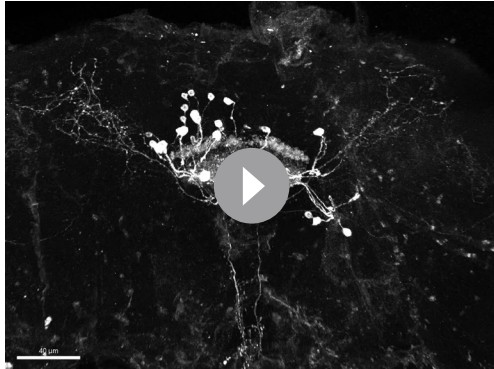

**Video 2.** Rotation of Imaris three-dimensional view of NPF-2A-LexA expression. Genotype: 20xUAS-flp;worniu-gal4,asense-gal80; lexAop-FRT-stop-FRT-myr::gfp x NPF-2A-LexA.

https://elifesciences.org/articles/105896/figures#video2

NPF and AstA); conversely, we also observe one neuropeptide expressed in multiple clusters (e.g. Ms, 2 clusters; Tk, 2 clusters; *Figure 6A*).

Next, we wanted to validate whether these neuropeptidergic cells indeed come from T2 lineages and target the CX. This required a method for specifically labeling of T2 peptidergic neurons, as antibody staining could be contaminated with T1-derived neurons. We developed a genetic approach to selectively visualize neuropeptide expression exclusively in neurons originating from T2 NB lineages. This technique involves driving FLP recombinase (FLP) under the control of the T2-specific driver *wor-Gal4,ase-Gal80*. The T2-specific FLP catalyzes the excision of a stop codon in the *lexAop-FRT-stop-FRT-myr::gfp* transgene, allowing existing neuropeptide-2A-LexA transgenes (*Deng et al., 2019*) to drive GFP expression specifically in neuropeptide-positive T2-derived neurons (*Figure 6B*). This method revealed the morphology of specific T2-derived peptidergic neurons.

We first assayed Ms+ neurons because *Wolff et al., 2025* reported that at least two FB neurons, FB4Z and FB5R express Ms and they target to two distinct FB layers. However, we found that T2-derived Ms-2A-LexA-expressing neurons project to multiple layers of the dorsal fan-shaped body and the entire ellipsoid body, suggesting an unknown class of Ms+ neurons targeting to EB and/or FB (*Figure 6C–C''*, *Video 1*). In our atlas, Ms is mainly expressed in clusters 157 and 160; however, we cannot distinguish the corresponding identity of each neuron.

Next, we assayed neurons in two clusters (100 and 128) that express the same two neuropeptides, AstA and NPF (*Figure 6D–E*). We wanted to determine if these clusters contain two cell types with similar gene expression, or a single cell type that co-expressed both neuropeptides. We used NPF-2A-lexA and AstA-2A-lexA to label each class of neurons. We found that they were generated at indistinguishable numbers of cells and projections: both had ~20 neurons projecting to a lateral domain of the ellipsoid body and the same two layers of the fan-shaped body (*Figure 6D–E*; *Video 2* and *Video 3*). Whether these T2-derived NPF and AstA neurons are indeed the same would require antibody validation.

We investigated whether specific transcription factors (TFs), or TF combinatorial codes, might correlate with the expression of cluster-defining neuropeptides, and thus provide candidates for a regulatory relationship where each cluster may have a TF code that drives transcription of a specific neuropeptides. To address this, we

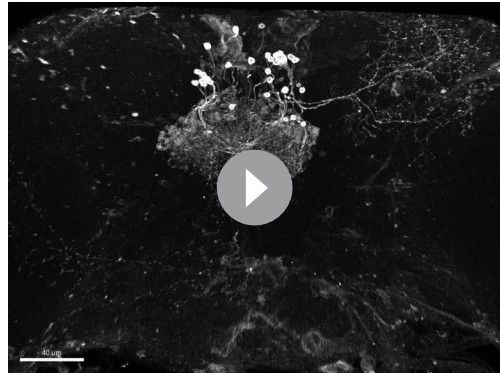

**Video 3.** Rotation of Imaris three-dimensional view of AstA-2A-LexA expression. Genotype: 20xUAS-flp;worniu-gal4,asense-gal80; lexAop-FRT-stop-FRT-myr::gfp x AstA-2A-LexA.

https://elifesciences.org/articles/105896/figures#video3

performed an unbiased correlation analysis of TF and neuropeptide expression patterns across all cells in our dataset to identify potential regulatory relationships. Interestingly, we found high correlations of multiple TFs to each neuropeptide (*Figure 6*, *Figure 6—figure supplement 1*, *Supplementary file 12*). Our data provide a stepping-stone to the analysis of upstream TF combinatorial codes that drive neuropeptide expression.

## Each neuron cluster consists of a unique combination of transcription factors

It is generally thought that neuronal diversity is generated by TF combinatorial codes that are uniquely expressed in neuronal subtypes, with an emphasis on homeodomain TFs (*Reilly et al., 2020*; *Sagner et al., 2021*). Thus, we assayed each class of TFs to determine how many were expressed in cluster-specific (i.e. neuron-specific) patterns. We found that most clusters expressed a unique TF combination (*Figure 7*). This is true for all zinc-finger TFs (*Figure 7A*) and helix-turn-helix TFs (*Figure 7B*). Nearly, all clusters expressed a unique combination of homeodomain TFs (*Figure 7C*), with only a few clusters sharing a homeodomain code (*Figure 7C'*). In contrast, basic domain, unspecified domain, and high mobility group TFs were more promiscuously expressed (*Figure 7D*). We conclude that zinc finger, helix-turn-helix, and homeodomain TFs are good candidates for forming unique combinatorial codes that should be tested for a role in generating neuronal diversity.

## Linking neurons to UMAP clusters

We sought to link well characterized CX neurons to their transcriptomes within the T2 atlas. We chose columnar neurons due to known cell type markers and genetic access. We used four complementary approaches to identify them, described below.

1. We assigned identities of neuronal clusters based on the expression profiles of neuropeptides and neurotransmitters (*Figure 6*) characterized in the literature (*Wolff et al., 2025*). We further examined these neurons of their expression of selective TFs and markers with antibody staining, including *bsh*, *cut*, *DIP-β*, *Eip93F*, *Imp*, *runt*, *Syp*, *toy*, and *zfh2*. The results are summarized in *Supplementary file 13*.

2. We used a published bulk RNAseq dataset to profile selective CX neurons and compared the expression profiles to our snRNA-seq dataset with a published algorithm (*Davis et al., 2020*; *Figure 8A*). We generated a heatmap based on the coefficient of the expression profiles to predict the identity of each cluster. We predicted that E-PG neurons were in cluster 75 and 95, P-EG neurons were in cluster 102, hDeltaK neurons were in cluster 100, FB6A were in cluster 162, and FB7A were in cluster 40 (*Figure 8A*). The prediction of E-PG neurons arising from cluster 75 and 95 was validated by expression of the transcription factor *dac* (*Dillon and Doe, 2024*); other neuron-cluster assignments await experimental validation.

3. We assayed the enhancers in highly specific split-Gal4 lines to determine if the associated genes were expressed in the same neuron subtype as the split-Gal4 line (*Figure 8B*). For example, the stable split line SS65380 is expressed in the FB2A neurons, and the enhancer DNA used to make the split lines are associated with the *wnt10* and *lmpt* genes. These two genes show high expression in cluster 149 (*Figure 8B*), allowing us to predict that the FB2A neurons are in cluster 149. A similar approach was used to generate candidate neuron-cluster predictions for the following neurons: FB8G, FB6H, hDeltaA, hDeltaD, hDeltaH, and FB1C (*Figure 8B*). These predictions can be used to focus experimental validation on a small group of neurons; however, the percentage of validated neurons-clusters awaits experimental analysis.

4. The fourth approach was based on using known or novel markers of CX neurons together with Gal4/LexA lines that are regulated by known gene enhancers. TFs known to be expressed by PF-R, P-EN and P-FN neurons are Toy and Runt, respectively (*Sullivan et al., 2019*). We assigned PF-R neurons to cluster 105 based on being Toy +Runt and being labeled by Gal4/LexA driven by the enhancer of *rho* gene (*Figure 8C*, left and middle columns). We assigned P-FN to clusters 38, 46, and 123 based on being Toy- Runt +and being labeled by split line SS00191 associated with *shakB* and *Pkc53E* genes. Lastly, we assigned P-EN neurons to clusters 66 and 130 by their expression of Toy- Runt +and labeled by *Gγ30A-lexA* (*Figure 8C*, left and middle columns). This approach identified novel TF markers for P-EN, PF-R and P-FN neurons (*Figure 8C*, right columns), which we validated by antibody staining (*Figure 8D–H*; *Dillon et al., 2024*). The results of cluster identity assignment are summarized in *Figure 9*; *Supplementary file 14*.

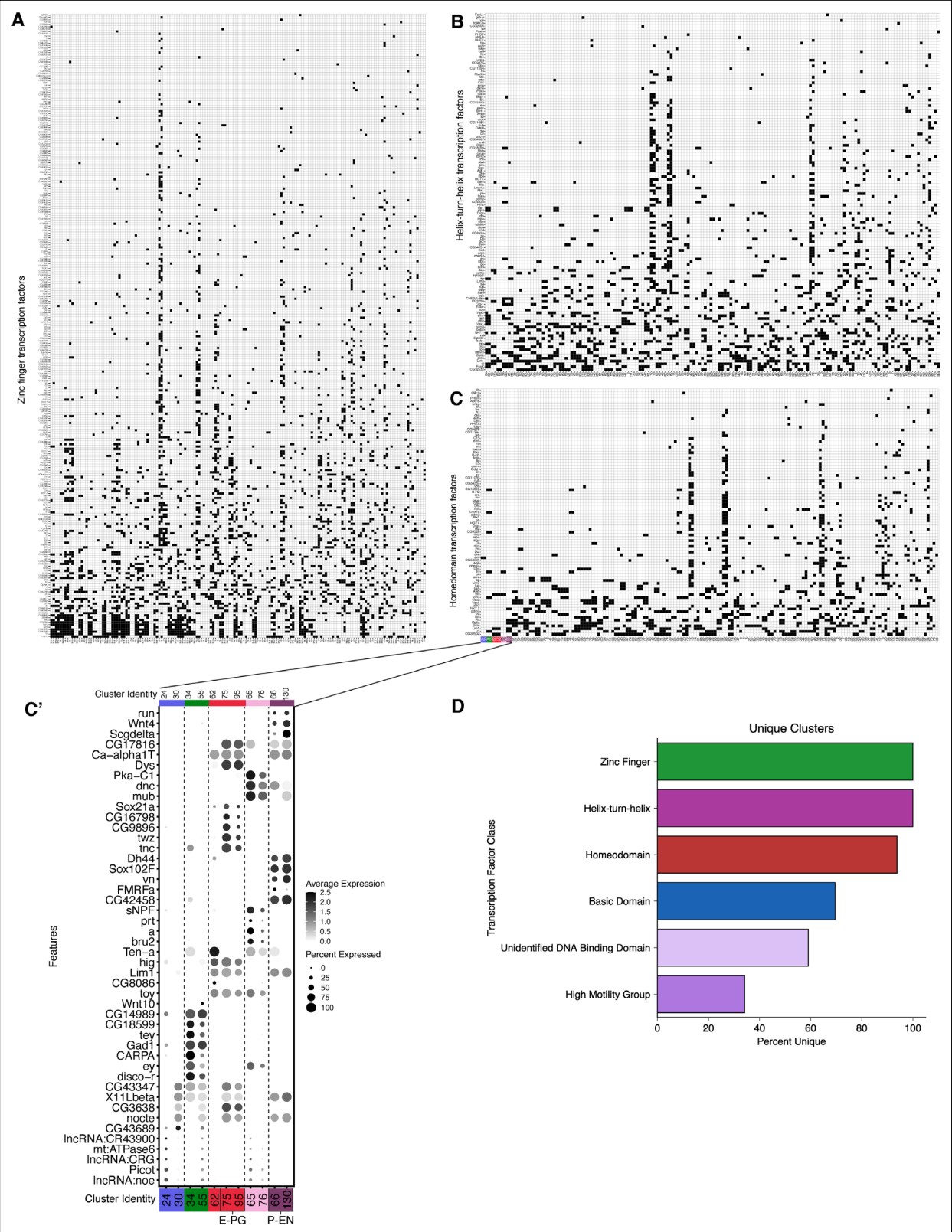

**Figure 7.** Combinations of transcription factor expression in T2-derived neurons. Binarized heatmaps of positive (**A**) zinc finger (**B**) helix-turn-helix (**C**) homeodomain TF markers in T2-derived neurons. Clusters are sorted by similarity based on Jaccard index scores. (**C'**) Top five marker genes for clusters which had non-unique combinations of homeodomain TF expression. Percentage of clusters with unique combinations of TF expression based on TF class zinc finger TFs = 100% (161 unique clusters), helix-turn-helix TFs = 100% (161 unique clusters), homeodomain TFs = 93.8% (150 unique clusters),

*Figure 7 continued on next page*

*Figure 7 continued*

basic domain TFs = 69.6% (112 unique clusters), unidentified DNA binding domain TFs = 59% (95 unique clusters), high motility group TFs = 34.2% (55 unique clusters).

The differential expression of TFs probably contributes to the morphological or connectivity diversity found within each class of CX neurons (*Hulse et al., 2021*; *Turner-Evans et al., 2020*; *Wolff and Rubin, 2018*; *Wolff et al., 2015*). We conclude that our T2 atlas can be used to link known neuron subtypes to their transcriptome (*Figure 9*). This is a necessary step in determining how cluster-specific gene expression regulates neuron-specific functional properties such as morphology, connectivity, and physiology.

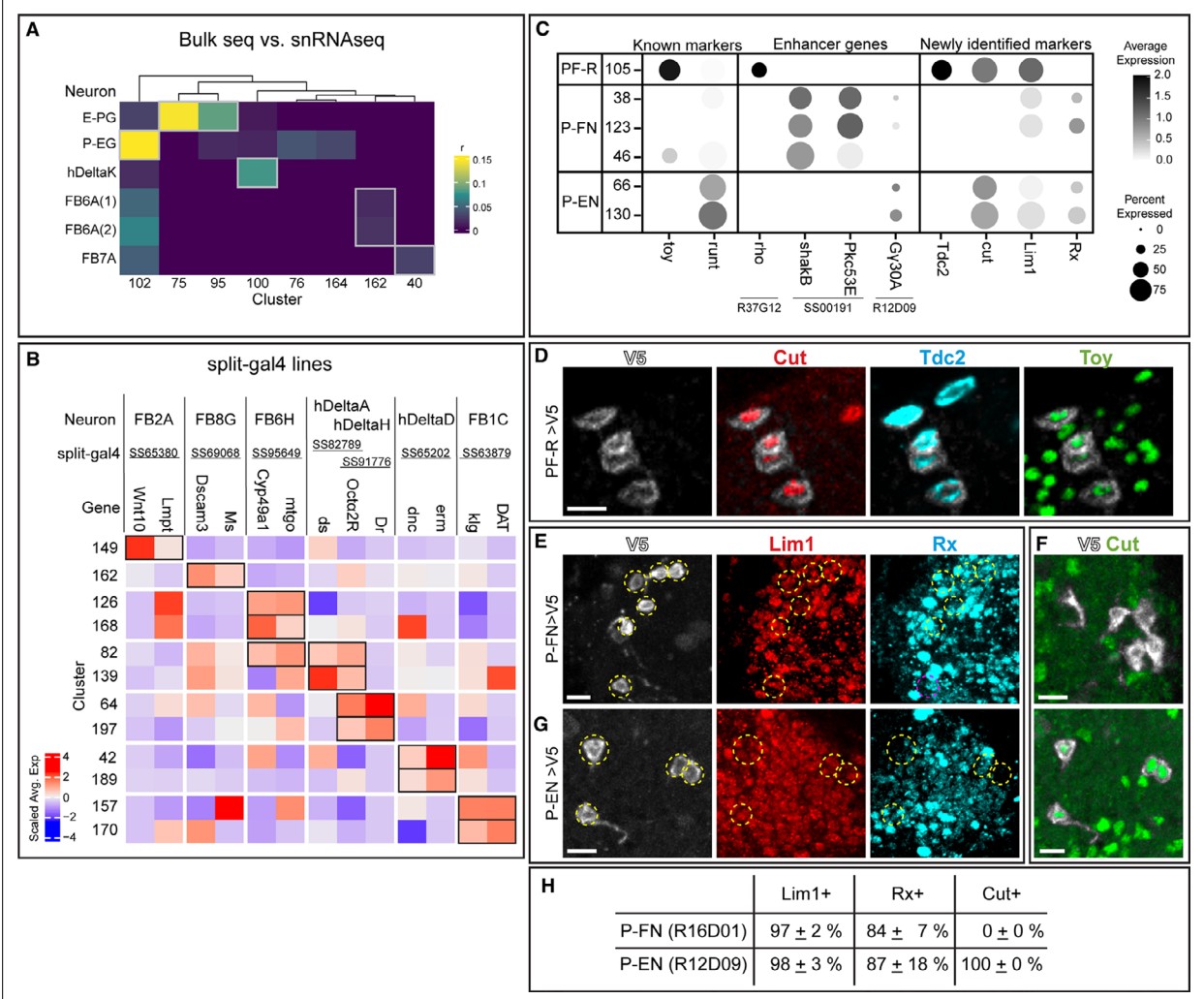

**Figure 8.** Mapping T2-derived neurons to UMAP clusters. (**A**) Heatmap of the coefficients (**r**) of expression profiles of single cell clusters (columns) to the expressions of known central complex neurons (rows) profiled with bulk RNA sequencing. The matching of the cluster to the central complex neurons is determined by the highest coefficient value (gray box) for each cluster. FB6A(1) and FB6A(2) are labeled and profiled with two different split-Gal4 drivers (*Wolff et al., 2025*). Cluster 75 and 95 identified as E-PG were verified in *Dillon et al., 2024*. (**B**) Scaled average expression of the enhancer genes of split-Gal4 drivers (columns) in each cluster (rows). The name of central complex neurons (CX) labeled by split-Gal4 drivers are listed at the top. The cluster identity was determined by the combination of positive scaled average expression of the enhancer genes (black boxes). (**C**) Dotplot of the known marker genes (*toy*, *runt*) expressed in central complex neurons (PF-R, P-FN, and P-EN), lexA-driver enhancer gene (*rho*, *Gγ30A*), split-Gal4-driver (SS00191) enhancer genes (*shakB*, *Pkc53E*), and newly identified marker genes. (**D**) PF-R (*R37G12-Gal4 UAS-V5*) neuronal cell bodies labeled with V5, and co-stained with marker genes boxed in (**C**). (**E,F**) PF-N (*R16D01-Gal4 UAS-V5*) neuronal cell bodies labeled with V5, and co-stained with marker genes boxed in (**C**). (**G**) PE-N (*R12D09-Gal4 UAS-V5*) neuronal cell bodies labeled with V5, and co-stained with marker genes boxed in (**C**). (**H**) Percentage of marker genes in PE-N and PF-N. Scale bar: 5 μm in all panels.

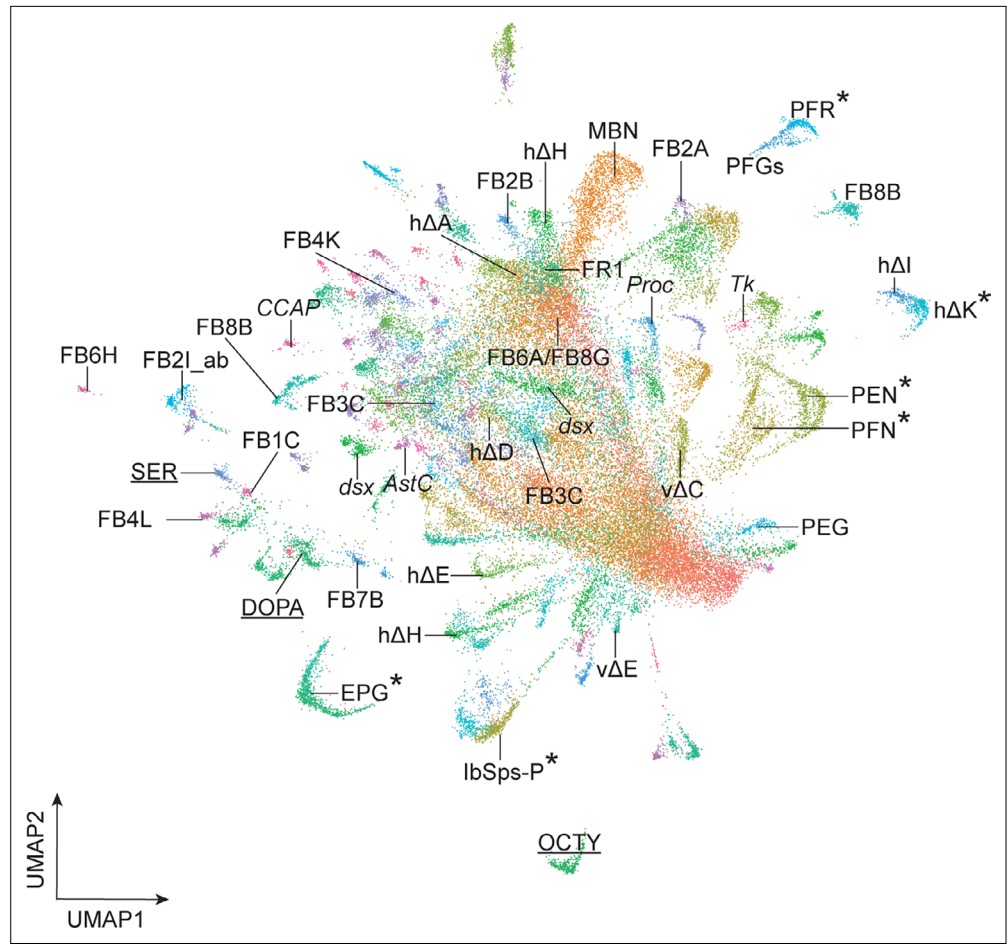

**Figure 9.** Summary. Asterisk, known neuron-cluster associations; others are predicted neuron-cluster associations. Underline, neurotransmitter expressing clusters. Abbreviations: *AstC, Allatostatin C+* neurons; *CCAP, Crustacean cardioactive peptide+* neurons; DOPA, dopaminergic neurons; *dsx, doublesex+* neurons; MBN, mushroom body neurons; OCTY, octopaminergic-tyraminergic neurons; *Proc,* Proctolin + neurons; *Tk,* Tachykinin+ neurons; SER, serotonergic neurons. E-PG, FB1C, FB2A, FB2B, FB2I_ab, FB3C, FB4K, FB4L, FB6A, FB6H, FB7B, FB8B, FB8G, FR1, hΔA, hΔD, hΔE, hΔH, hΔI, hΔK, lbSps-P, P-EG, P-EN, P-FGs, P-FN, PF-R, vΔC and vΔE are the names of central complex neurons (**Hulse et al., 2021**).

## Discussion

We used a genetic approach to express RFP in the progeny of T2 NBs in the adult central brain. We excluded optic lobes by manual dissection, and thus our data are similar to recent scRNA-seq data from the central brain (**Croset et al., 2018**; **Davie et al., 2018**; **Li et al., 2022**). Similarities include neurons expressing predominantly single fast-acting neurotransmitters (discussed below). Both datasets show major classes of neurons including MB neurons, olfactory projection neurons, clock neurons, Poxn+ neurons, serotonergic neurons, dopaminergic neurons, octopaminergic neurons, corazonergic neurons, and hemocytes. The MB neurons that appear in the T2-derived progeny may be due to off-target expression of our Gal4 driver at stages we have not assayed, or more likely, due to ambient RNA released during the dissociation and sorting steps.

Interestingly, we observed fewer distinct neuronal clusters in the T1-derived population (114 clusters; *Figure 2—figure supplement 1*) than in the T2-derived neuronal population (161 clusters; *Figure 2*). This could be due to the T1-derived progeny containing a large population of transcriptionally similar Kenyon cells of the MB, whereas T2-derived clusters contain fewer, but transcriptionally more diverse, cells than T1-derived progeny. It would be interesting to know whether T1 NB lineages are generally less diverse, or whether the Kenyon cells are exceptional in their lack of transcriptional

diversity, despite the presence of protein gradient in young vs. old neurons (*Liu et al., 2015*). Our results are consistent with lineage analysis of T2 NBs, which make distinct neurons across the temporal axis of T2 NBs, plus unique neurons across the INP temporal axis (*Doe, 2017*; *Ito et al., 2013*; *Sullivan et al., 2019*; *Wang et al., 2014*; *Yang et al., 2013*).

### Neurotransmitters and neuropeptides

We observed that most fast-acting neurotransmitters are uniquely expressed in neuronal populations, but we also observed a smaller group of neurons that express two or three neurotransmitters. This includes excitatory and inhibitory neurons, for example 2.9% of all neurons express genes necessary for cholinergic and GABAergic neurotransmitters (*Figure 3*). Co-expression of neurotransmitters has been reported in other systems (*Granger et al., 2020*; *Granger et al., 2023*; *Lozovaya et al., 2018*; *Meye et al., 2016*; *Saunders et al., 2015*; *Shabel et al., 2014*; *Spitzer, 2015*; *Takács et al., 2018*) and it may be biologically relevant in the T2-derived neurons.

Neuropeptides are profoundly important for a vast array of behaviors (*Schoofs et al., 2017*), and thus our mapping of neuropeptide expression to distinct neuronal subtypes may facilitate a functional analysis linking neuron identity, neuropeptide expression, and behavior.

### Glial identities

We identified six classes of glia in the T1+T2 glial atlas that reflects previous scRNAseq datasets (*Figure 1E–F*; *Konstantinides et al., 2018*; *Lago-Baldaia et al., 2023*). Similar to a recent glial cell atlas (*Lago-Baldaia et al., 2023*), we found glial subtypes like astrocytes, ensheathing, and subperi-neurial glia mapped to several clusters (*Figure 1E–F*). It remains unclear if these clusters with the same cell type annotation represent distinct glial identities or different transcriptional states within these populations. Interestingly, we detected a small population of *vkg+* surface glia specific to the whole atlas (T1+T2) and not the T2 glia atlas. The lack of differential gene expression between T1- and T2-derived glia suggests these cell types are similar despite different developmental origins. It will be for future work to understand how different progenitors produce similar glial fates.

Similar to the T1+T2 glial atlas, our T2 atlas captured the known glial subtypes found in other scRNAseq datasets (*Figure 4*; *Konstantinides et al., 2018*; *Lago-Baldaia et al., 2023*). The T2 glia atlas contains more clusters than the T1+T2 atlas that may be due to the nearly 4x the number of glia captured (T1+T2 atlas 3,409 nuclei; T2 atlas – 12,315 nuclei). Interestingly, we identified the astrocyte-like glia of the central brain previously described (*Awasaki et al., 2008*). Additionally, we captured chiasm glia that have been shown to be derived from the T2 lineage DL1 and migrate into the optic lobe (*Viktorin et al., 2013*). These nuclei could represent either chiasm glia that did not migrate out of the central brain or optic lobe tissue that was not completely removed during dissections. It will be interesting to investigate how these T2-derived migratory chiasm glia compared to the optic-lobe derived chiasm glia, because of their different developmental origins.

### Sex differences found in the adult neurons and glia

The T2 glia and neuron atlases contained clusters with disproportionate enrichment of female and male nuclei when normalized for sample input. We found the expected differential expression of yolk protein transcripts (*yp1, yp2, yp3*) enriched in female nuclei and the long non-coding RNAs *rox1/2* and *fru* enriched in male nuclei (*Amrein and Axel, 1997*; *Meller et al., 1997*; *Ryner et al., 1996*). Inter-estingly, we found *dsx* to be enriched in both glial and neuronal female nuclei. Surprisingly, we found additional female and male-specific gene enriched in both neurons and glia including *ATPsyndelta* and undescribed, computationally predicted genes (CGs; *Figure 5*). It remains to be determined if these genes are driving sex-specific differences within glial and neuronal subtypes. These genes may reflect sex-specific differences in the adult central brain and may provide insight into how behavioral circuits are linked to sex-specific behaviors. Future work should aim to characterize and test these genes for functional roles.

### TF codes

Our atlas showed zinc finger TFs and homeodomain TFs as forming combinatorial codes specific for distinct cell types. This raises the possibility that these TF combinations determine neuronal func-tional properties, by establishing and maintaining neuronal identities. This is consistent with data

from *C. elegans* (*Hobert, 2021*; *Reilly et al., 2020*), mammalian spinal cord (*Briscoe et al., 2000*; *Sagner et al., 2021*), *Drosophila* optic lobe (*Holguera and Desplan, 2018*; *Konstantinides et al., 2018*), leg motor neurons (*Baek et al., 2013*; *Enriquez et al., 2015*), and VNC (*Soffers et al., 2024*). Interestingly, we found more zinc finger TFs expressed in unique combinatorial codes compared to homeodomain TFs; this may be a caution to focusing too narrowly on homeodomain TFs function. We note that our results are mostly correlative and await functional analysis of the TF combinatorial codes found in our dataset. It is also possible that the current sparse TF expression patterns are due, in part, to false negative results due to low read depths. This can be resolved by increasing read depth or by performing functional assays.

## Mapping neurons to clusters

In this study, we used several complementary approaches to map identified neurons to their transcriptomic UMAP cluster. First, we used bulk-seq data from specific neurons with our atlas to match neuron to cluster. Second, we used enhancer elements from neuron-specific split Gal4 lines to link enhancer-associated genes to each hemi-driver with common expression in a cluster. Third, we used known or novel TFs and other molecular markers that are expressed by a cluster as an entry point to look for other cluster specific genes. Each of these methods provides candidate neuron-cluster association, which requires experimental validation. It is not clear which method is more robust at identifying functional neuron-cluster associations. Nevertheless, we have validated a number of neuron-cluster associations using the third approach: beginning with a single gene-cluster correlation. In the future, our data can be used in two pipelines for linking neuron subtype to transcriptome. First, start with at least one validated marker for a cluster, then search all labeled clusters for candidate TFs that co-express with the validated marker, followed by validation via RNA *in situ* (or antibodies). Second, we can collect neurons of interest with FACS or antibody selection (*Davis et al., 2020*), and perform bulk RNA sequencing, followed by looking for clusters enriched for the TFs found in bulk sequencing data, and then validation via RNA *in situ* (or antibodies). Notably, the second pipeline requires no prior markers beyond the Gal4 or LexA line expressed in single neuronal populations. Both pipelines result in a transcriptome that can be used to identify (a) TFs for their role in neuronal specification and/or maintenance, (b) cell surface molecules that may regulate neuronal morphology and/or connectivity, or (c) functionally relevant genes encoding ion channels, neuropeptides, receptors, and signaling pathways.

# Materials and methods

## Key resources table

| Reagent type (species) or resource | Designation | Source or reference | Identifiers | Additional information |
|---|---|---|---|---|
| Genetic reagent (*Drosophila melanogaster*) | *20xUAS-FLPG5.PEST;worniu-gal4,asense-gal80; Act5c(FRT.CD2)gal4* | *Syed et al., 2017* | | Type II lineage immortalization |
| Genetic reagent (*Drosophila melanogaster*) | *20xUAS-FLPG5.PEST;worniu-gal4,asense-gal80; lexAop(FRT.stop)-mCD8:GFP* | This work | | Label type II derived lexA + cells |
| Genetic reagent (*Drosophila melanogaster*) | *TI{2A-lexA::GAD}AstA [2A-lexA]/TM3,Sb[1]* | BDSC | RRID:BDSC_84356 | |
| Genetic reagent (*Drosophila melanogaster*) | *TI{2A-lexA::GAD}Ms [2A-lexA]/TM3,Sb[1]* | BDSC | RRID:BDSC_84403 | |
| Genetic reagent (*Drosophila melanogaster*) | *TI{2A-lexA::GAD}NPF [2A-lexA]/TM3,Sb[1]* | BDSC | RRID:BDSC_84422 | |
| Genetic reagent (*Drosophila melanogaster*) | *UAS-RedStinger* | BDSC | RRID:BDSC_8545 | |
| Genetic reagent (*Drosophila melanogaster*) | *UAS-unc84-2xGFP* | *Henry et al., 2012* | | |
| Genetic reagent (*Drosophila melanogaster*) | *GMR12D09-lexA/CyO* | BDSC | RRID:BDSC_54419 | P-EN |
| Genetic reagent (*Drosophila melanogaster*) | *GMR16D01-lexA* | BDSC | RRID:BDSC_52503 | P-FN |

*Continued on next page*

*Continued*

| Reagent type (species) or resource | Designation | Source or reference | Identifiers | Additional information |
|---|---|---|---|---|
| Genetic reagent (*Drosophila melanogaster*) | *GMR37G12-lexA* | BDSC | RRID:BDSC_52765 | PF-R |
| Genetic reagent (*Drosophila melanogaster*) | *13xLexAop2-IVS-myr::smGdP-V5* | BDSC | RRID:BDSC_62215 | |
| Antibody | mouse anti-Cut 2B10, monoclonal | DSHB | RRID:AB_528186 | 2 µg/mL |
| Antibody | guinea pig anti-DIP-β, polyclonal | *Xu et al., 2024* | | 1:300 |
| Antibody | guinea pig anti-E93, polyclonal | *Syed et al., 2017* | | 1:500 |
| Antibody | rabbit anti-Imp, polyclonal | *Syed et al., 2017* | | 1:500 |
| Antibody | rabbit anti-Lim1, polyclonal | Desplan, New York University | | 1:500 |
| Antibody | guinea pig anti-Runt, polyclonal | *Sullivan et al., 2019* | | 1:1000 |
| Antibody | guinea pig anti-Rx, polyclonal | Desplan, New York University | | 1:500 |
| Antibody | rabbit anti-Syp, polyclonal | *Syed et al., 2017* | | 1:500 |
| Antibody | rabbit anti-Toy, polyclonal | *Sullivan et al., 2019* | | 1:1000 |
| Antibody | chicken anti-V5, polyclonal | Fortis Life Sciences, Waltham, MA | | 1 µg/mL |
| Antibody | rat anti-Zfh2, olyclonal | *Tran et al., 2010* | | 1:200 |
| Antibody | DyLight405, Alexa Fluor 488, Rhodamine Red-X (RRX), or Alexa Fluor 647 conjugated donkey whole IgG, polyclonals | Jackson Immuno Research Laboratories Inc, West Grove, PA | | 5 µg/mL |
| Commercial kit | Evercode WT | Parse Bioscience | | |
| Commercial kit | PIPseq 3' Single Cell RNA T20 kit | Fluent BioSciences | | |
| Software, algorithm | R Studio | Posit Software | | https://posit.co/products/open-source/rstudio/ |
| Software, algorithm | Seurat | Rahul Satija, New York University | | https://satijalab.org/seurat/ |
| Software, algorithm | ComplexHeatmap | *Gu, 2022* | | https://github.com/jokergoo/ComplexHeatmap (*Gu, 2025*) |

## Single nuclei isolation, library preparation, and sequencing

We first used the split-pool method (Parse Bioscience, Seattle, WA, USA) to barcode RNAs from whole brain to generate a library which includes both T1 and T2 progeny. This library, T1+T2, was used for analysis in *Figures 1 and 5*. To increase the number of T2 nuclei for snRNAseq, we labeled T2 nuclei by crossing *20xUAS-FLPG5.PEST;worniu-Gal4,asense-Gal80; Act5c(FRT.CD2)Gal4* to *UAS-RedStinger* (RFP) or *UAS-unc84-2xGFP* (GFP) flies. The adult flies were aged for 1 week at 25 °C before dissection. Equal amounts of male and female central brains (excluding optic lobes) were dissected at room temperature within 1 hr. The samples were flash-frozen in liquid nitrogen and stored separately at –80 °C. Dissociation of nuclei from the frozen, dissected brains was performed according to published protocols (*McLaughlin et al., 2022*). RFP +or GFP +nuclei were collected by sorting dissociated nuclei with SONY-SH800 with 100 µm chip. We performed three rounds of sorting and snRNAseq. In the first round, we pooled male and female brains together to select GFP

+ nuclei and used particle-templated instant partitions to capture single nuclei to generate cDNA library (Fluent BioSciences, Waterton, MA). In the second round, RFP +nuclei from male and female were pooled together. In the third round, RFP +nuclei from male and female brains were collected separately. The split-pool method was then used to generate barcoded cDNA libraries from each individual nucleus from the second and third rounds. All libraries were sequenced with pair-ends reads 150 bp on Illumina Novaseq 6000 (University of Oregon's Genomics and Cell Characterization Core Facility).

### snRNA-seq analysis

Our bioinformatic analysis was performed using pipeline from Fluent BioSciences, Parse Bioscience, and the Seurat R package (*Hao et al., 2024*; *Satija et al., 2015*). Briefly, pipelines from Fluent BioSciences and Parse Biosciences were used to perform demultiplexing, alignment, filtering, counting of barcodes and UMIs with an output being a cell-by-genes matrix of counts. We aligned our sequences to a custom reference genome by adding *flpD5* (RRID:Addgene_32132), *redstinger* (RRID:Addgene_46165), and *unc84sfGFP* (RRID:Addgene_46023; +SV40 tail) sequences and annotations to the *Drosophila* genome release BDGP6.32.109.To further ensure that only high-quality cells were retained, we removed any cells with fewer than 200 genes or more than 2500 genes expressed and more than 5% mitochondrial RNA. For the T2 atlas, snRNA-seq data from three rounds of sorting and barcoding were integrated in Seurat with Anchor-based RPCA integration to generate an integrated dataset, and the downstream analysis was performed with the default parameters. The UMAP was generated with resolution 12. The integrated dataset was used for analyses.

### Glia analyses

The T1+T2 glia atlas dataset was derived from the T1+T2 *repo+* glial clusters and re-clustered to represent the glial subtypes. The T2 glia atlas was derived from the T2 *repo+* glial clusters and re-clustered to represent the glial subtypes. Cell identity was determined by the validated markers for glia shown to differentially expressed between the clusters. Identities were assigned based on expression of these validated markers. Overlapping identities were assigned if multiple subtype markers were expressed within a single cluster.

### Sex differences

We determined sex-biased clusters within the T2 glia and neuron atlases by identifying clusters that were disproportionate after normalizing the number of inputs for 'female' and 'male' samples respectively. The 'female and male' mixed samples were excluded from the analyses as we could not differentiate the sex origin of these nuclei. To determine differential gene expression between females and males, we pseudo bulked nuclei by aggregating the snRNAseq for comparison between sex and clusters. Heatmaps were generated in Seurat using the Scillus package (https://github.com/xmc811/Scillus; *Xu, 2021*) to display the heatmap. Both male and female *Drosophila melanogaster* were used for input into the RNA-seq pipeline.

### Transcription factor combinatorial analysis

We used the Seurat function FindAllMarkers to find positively differentially expressed TFs in all clusters. We then removed TFs which were not significantly differentially expressed. We gave a value of 1 to any TF that was found to be a positive marker in each cluster and a value of 0 in clusters which it was not a positive marker. Next, we found the number of unique combinations of markers for six classes (zinc finger, helix-turn-helix, homeodomain, basic domain, unidentified DNA binding domain, and high motility group) of TFs. The Jaccard index between each cluster was then calculated and clusters were sorted from most to least similar. The python packages Matplotlib and Seaborn were then used to generate heatmaps.

## Immunohistochemistry and imaging

Standard methods were used for adult brain staining (*Sullivan et al., 2019*). The antibody stained brains were mounted in DPX (https://www.janelia.org/project-team/flylight/protocols) on poly-L-Lysine coated coverslip (Corning, Glendale, AZ) and imaged with Zeiss confocal LSM800 with software Zen.

## Contact for reagent and resource sharing

Further information and requests for resources and reagents should be directed to and will be fulfilled by the corresponding author Chris Doe (cdoe@uoregon.edu).

## Figure production

We used Imaris (Bitplane, Abingdon, UK) for confocal image processing, and Illustrator (Adobe, San Jose, CA) to assemble figures.

## Acknowledgements

We thank Jason Carriere, and Peter Newstein for comments on the manuscript. We thank the Desplan lab (NYU) for reagents. We thank Jason Carriere (University of Oregon Genomics and Cell Characterization Core Facility) for technical assistances in cDNA library preparation and sequencing. Flies were obtained from the Bloomington Stock Center.

## Additional information

### Funding

| Funder | Grant reference number | Author |
|---|---|---|
| Howard Hughes Medical Institute | | Derek G Epiney<br>Gonzalo Morales Chaya<br>Noah R Dillon<br>Sen-Lin Lai<br>Chris Q Doe |
| National Institutes of Health | 5-T32-HD07348 | Noah R Dillon |
| National Institutes of Health | 5-T32-GM149387 | Gonzalo Morales Chaya |
| National Institutes of Health | HD27056 | Derek G Epiney<br>Gonzalo Morales Chaya<br>Noah R Dillon<br>Chris Q Doe |

The funders had no role in study design, data collection and interpretation, or the decision to submit the work for publication.

### Author contributions

Derek G Epiney, Conceptualization, Resources, Data curation, Formal analysis, Investigation, Methodology, Writing – review and editing; Gonzalo Morales Chaya, Conceptualization, Resources, Data curation, Software, Formal analysis, Investigation, Visualization, Methodology, Writing – original draft, Writing – review and editing; Noah R Dillon, Conceptualization, Formal analysis, Validation, Investigation, Visualization, Methodology, Writing – original draft, Writing – review and editing; Sen-Lin Lai, Conceptualization, Resources, Data curation, Software, Formal analysis, Supervision, Validation, Investigation, Visualization, Methodology, Writing – original draft, Project administration, Writing – review and editing; Chris Q Doe, Conceptualization, Formal analysis, Supervision, Funding acquisition, Validation, Writing – original draft, Project administration, Writing – review and editing

### Author ORCIDs

Sen-Lin Lai ⬥ https://orcid.org/0000-0002-7531-283X

Chris Q Doe  https://orcid.org/0000-0001-5980-8029

## Ethics

No vertebrate animals were used in this study.

Reviewer #1 (Public review): https://doi.org/10.7554/eLife.105896.3.sa1
Reviewer #2 (Public review): https://doi.org/10.7554/eLife.105896.3.sa2
Author response https://doi.org/10.7554/eLife.105896.3.sa3

---

## Additional files

### Supplementary files

Supplementary file 1. Cluster-defining genes and cell identity of all clusters identified by Seurat function FindAllMarkers.

Supplementary file 2. Marker genes used to identify cell types.

Supplementary file 3. Marker genes used to identify glial cell types.

Supplementary file 4. Cluster-defining genes and glial cell types from all glial clusters identified by Seurat function FindAllMarkers.

Supplementary file 5. Differential expression (DE) analysis of glial marker genes in T1- vs T2-derived glia with Seurat function differential expression testing.

Supplementary file 6. Cluster-defining genes of T1-derived neuronal clusters identified by Seurat function FindAllMarkers.

Supplementary file 7. Cluster-defining genes of T2-derived neuronal clusters identified by Seurat function FindAllMarkers.

Supplementary file 8. Top 10 most enriched genes for each cluster extracted from *Supplementary file 6* by scCustomize function Extract_Top_Markers (*Marsh, 2024*; scCustomize: Custom Visualizations & Functions for Streamlined Analyses of Single Cell Sequencing; https://zenodo.org/records/14529706 RRID:SCR_024675).

Supplementary file 9. Cluster-defining genes of T2-derived glial clusters identified by Seurat function FindAllMarkers.

Supplementary file 10. Differential expression (DE) analysis of all genes in male vs female T2-derived glia with Seurat function differential expression testing.

Supplementary file 11. Differential expression (DE) analysis of all genes in male vs female T2-derived neurons with Seurat function differential expression testing.

Supplementary file 12. Coefficients of expression levels between transcription factors and neuropeptides in each individual nuclei.

Supplementary file 13. Expression profiles of selected genes by antibody staining in central complex neurons and predicted cluster.

Supplementary file 14. Summarized cluster identity of T2-derived neurons.

MDAR checklist

### Data availability

Sequencing data have been deposited in GEO under accession code GSE294658. All data and code can be found at https://github.com/dgepiney/2023_Doe_Drosophila_Central_Brain_RNAseq (copy archived at *Epiney, 2025*).

The following dataset was generated:

| Author(s) | Year | Dataset title | Dataset URL | Database and Identifier |
|---|---|---|---|---|
| Epiney D, Chaya GNM, Dillon NR, Lai S-L, Doe CQ | 2025 | Transcriptional complexity in the insect central complex: single nuclei RNA-sequencing of adult brain neurons derived from type 2 neuroblasts | https://www.ncbi.nlm.nih.gov/geo/query/acc.cgi?acc=GSE294658 | NCBI Gene Expression Omnibus, GSE294658 |

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
