## [Editor Report · eLife Assessment]

This **important** study offers a molecular characterization of neurons and glia in the adult nervous system of the fruit fly *Drosophila melanogaster*. The study focuses on the progeny of a specific set of neural stem cells that contribute to the central complex, a conserved brain region that plays key roles in sensorimotor integration. The data are **convincing** and collected using validated methodology, generating an invaluable resource for future studies. The study will be of interest to developmental neurobiologists.

---

## [Referee Report · Reviewer #1 (Public review)]

Summary:

Epiney et al. use single-nuclei RNA sequencing (snRNA-seq) to characterize the lineage of Type-2 (T2) neuroblasts (NBs) in the adult *Drosophila* brain. To isolate cells born from T2 NBs, the authors used a genetic tool that specifically allows the permanent labeling of T2-derived cell types, which are then FAC-sorted for snRNA-seq. This effective labeling approach also allows them to compare the isolated T2 lineage cells with T1-derived cell types by a simple exclusion method. The authors begin by describing a transcriptomic atlas for all T1 and T2-derived neuronal and glia clusters, reporting that the T2-derived lineage comprises 161 neuronal clusters, in contrast to the T1 lineage which comprises 114 of them. The authors then use the expression of VAChT, VGlut, Gad1, Tbh, Ple, SerT, and Tdc2 to show that T2 neuroblasts generate all major neuron classes of fast-acting neurotransmitters. Strikingly, they show that a subset of glia and neuronal clusters have disproportionate enrichment in males or females, suggesting that T2 neuroblasts generate sex-biased cell types. The authors then proceed to characterize neuropeptide expression across T2-derived neuronal clusters and argue that the same neuropeptide can be expressed across different cell types, while similar cell types can express distinct neuropeptides. The functional implication of both observations, however, remains to be tested. Furthermore, the authors describe combinatorial transcription factor (TF) codes that are correlated with neuropeptide expression for T2-derived neurons along with an overall TF code for all T2-derived cell types, both of which will serve as an important starting point for future investigations. Finally, the authors map well-studied neuronal types of the central complex to the clusters of their T2-derived snRNA-seq dataset. They use known marker combinations, bulk RNA-seq data and highly specific split-Gal4 driver lines to annotate their T2-derived atlas, establishing a comprehensive transcriptomic atlas that would guide future studies in this field.

Strengths:

This study provides an in-depth transcriptomic characterization of neurons and glia derived from Type-2 neuroblast lineages. The results of this manuscript offer several future directions to investigate the mechanisms of diversifying neuronal identity. The datasets of T1-derived and T2-derived cells will pave the way for studies focused on the functional analysis of combinatorial TF codes specifying cell identity, sex-based differences in neurogenesis and gliogenesis, the relationship between neuropeptide (co)expression and cell identity, and the differential contributions of distinct progenitor populations to the same cell type.

Weaknesses:

The study presents several important observations based on the characterization of Type II neuroblast-derived lineages. However, a mechanistic insight is missing for most observations. The idea that there is a sex-specific bias to certain T2-derived neurons and glial clusters is quite interesting, however, the functional significance of this observation is not tested or discussed extensively. Finally, the authors do not show whether the combinatorial TF code is indeed necessary for neuropeptide expression or if this is just a correlation due to cell identity being defined by TFs. Functional knockdown of some candidate TFs for a subset of neuropeptide-expressing cells would have been helpful in this case.

Comments on revisions:

The authors have addressed my recommendations.

---

## [Referee Report · Reviewer #2 (Public review)]

In this manuscript, Epiney et al., present a single-nucleus sequencing analysis of *Drosophila* adult central brain neurons and glia. By employing an ingenious permanent labeling technique, they trace the progeny of T2 neuroblasts, which play a key role in the formation of the central complex. This transcriptomic dataset is poised to become a valuable resource for future research on neurogenesis, neuron morphology, and behavior.

The authors further delve into this dataset with several analyses, including the characterization of neurotransmitter expression profiles in T2-derived neurons. While some of the bioinformatic analyses are preliminary, they would benefit from additional experimental validation in future studies.

Comments on revisions:

We appreciate the authors' efforts to address some of the comments. While these revisions have improved the clarity of certain sections, some of the larger concerns remain unaddressed. Specifically, the manuscript still lacks the additional analyses that would allow for more specific conclusions, rather than the general observations currently presented. Although the revisions have certainly made the text clearer, the core issue of needing more detailed analysis to draw more concrete conclusions still stands.

---

## [Author Response]

The following is the authors’ response to the original reviews

**Public Reviews:**

**Reviewer #1 (Public review):**
Summary:Epiney et al. use single-nuclei RNA sequencing (snRNA-seq) to characterize the lineage of Type-2 (T2) neuroblasts (NBs) in the adult *Drosophila* brain. To isolate cells born from T2 NBs, the authors used a genetic tool that specifically allows the permanent labeling of T2-derived cell types, which are then FAC-sorted for snRNA-seq. This effective labeling approach also allows them to compare the isolated T2 lineage cells with T1-derived cell types by a simple exclusion method. The authors begin by describing a transcriptomic atlas for all T1 and T2-derived neuronal and glia clusters, reporting that the T2-derived lineage comprises 161 neuronal clusters, in contrast to the T1 lineage which comprises 114 of them. The authors then use the expression of VAChT, VGlut, Gad1, Tbh, Ple, SerT, and Tdc2 to show that T2 neuroblasts generate all major neuron classes of fast-acting neurotransmitters. Strikingly, they show that a subset of glia and neuronal clusters have disproportionate enrichment in males or females, suggesting that T2 neuroblasts generate sex-biased cell types. The authors then proceed to characterize neuropeptide expression across T2-derived neuronal clusters and argue that the same neuropeptide can be expressed across different cell types, while similar cell types can express distinct neuropeptides. The functional implication of both observations, however, remains to be tested. Furthermore, the authors describe combinatorial transcription factor (TF) codes that are correlated with neuropeptide expression for T2-derived neurons along with an overall TF code for all T2-derived cell types, both of which will serve as an important starting point for future investigations. Finally, the authors map well-studied neuronal types of the central complex to the clusters of their T2-derived snRNA-seq dataset. They use known marker combinations, bulk RNA-seq data and highly specific split-Gal4 driver lines to annotate their T2-derived atlas, establishing a comprehensive transcriptomic atlas that would guide future studies in this field.

Thanks for the clear and accurate summary of our findings.

Strengths:This study provides an in-depth transcriptomic characterization of neurons and glia derived from Type-2 neuroblast lineages. The results of this manuscript offer several future directions to investigate the mechanisms of diversifying neuronal identity. The datasets of T1-derived and T2-derived cells will pave the way for studies focused on the functional analysis of combinatorial TF codes specifying cell identity, sex-based differences in neurogenesis and gliogenesis, the relationship between neuropeptide (co)expression and cell identity, and the differential contributions of distinct progenitor populations to the same cell type.

Thank you for the positive comments.

Weaknesses:The study presents several important observations based on the characterization of Type II neuroblast-derived lineages. However, a mechanistic insight is missing for most observations. The idea that there is a sex-specific bias to certain T2-derived neurons and glial clusters is quite interesting, however, the functional significance of this observation is not tested or discussed extensively. Finally, the authors do not show whether the combinatorial TF code is indeed necessary for neuropeptide expression or if this is just a correlation due to cell identity being defined by TFs. Functional knockdown of some candidate TFs for a subset of neuropeptide-expressing cells would have been helpful in this case.

We agree that we do not provide mechanistic or functional insights. Our goal was to produce hypothesis generating datasets for our lab and others to use to direct functional or mechanistic studies.

**Reviewer #2 (Public review):**
In this manuscript, Epiney et al., present a single-nucleus sequencing analysis of *Drosophila* adult central brain neurons and glia. By employing an ingenious permanent labeling technique, they trace the progeny of T2 neuroblasts, which play a key role in the formation of the central complex. This transcriptomic dataset is poised to become a valuable resource for future research on neurogenesis, neuron morphology, and behavior.

Thank you for the positive comments.

The authors further delve into this dataset with several analyses, including the characterization of neurotransmitter expression profiles in T2-derived neurons. While some of the bioinformatic analyses are preliminary, they would benefit from additional experimental validation in future studies.

Thank you for the positive comments. We too hope that future research will benefit from this dataset.

**Reviewer #1 (Recommendations for the authors):**
Major points(1) In Figures 1E and 4A, the T1 and T2 glia subsets reveal sub-clusters for several cell types as seen by the distribution of points on the UMAP. This observation is never validated or discussed. Do these sub-clusters represent true differences in identities or are they artifacts of the single-nucleus preparation? For Figure 1E, it is not clear whether specific sub-clusters (see Ensheathing-4 vs Ensheathing-5 and Astrocyte-2 vs. Astrocyte-6) are differentially enriched between the T1 and T2 lineages. The existence of these sub-clusters must be discussed or dismissed.

We agree that this needs to be addressed more clearly in the manuscript and have made text changes in the Results and Discussion sections to clarify. We note that a recent glial cell atlas (Lago-Baldaia et al., 2023: PMID: 37862379) of the developing fly VNC and optic lobes found sub-clusters that mapped to the same subtype annotations. Interestingly, Lago-Baldaia and colleagues found that the transcriptional diversity of glia cell types did not match the morphological diversity of glia validated *in vivo*. See text changes below.

Lines 131-133: “Similar to a previous glial cell atlas (Lago-Baldaia et al., 2023) we found some glial subtypes (astrocytes, ensheathing, and subperineurial) mapped to multiple clusters (Figure 1E, 1F).”

Lines 206-208: “In line with our T1+T2 atlas and previous glia cell atlas (Lago-Baldaia et al., 2023), some subtypes mapped to several subclusters including ensheathing, astrocytes, and chiasm (Figure 4A-B).”

Lines 397-401: “Similar to a recent glial cell atlas (Lago-Baldaia et al., 2023), we found glial subtypes like astrocytes, ensheathing, and subperineurial glia mapped to several sub-clusters (Figure 1E-F). It remains unclear if these sub-clusters with the same cell type annotation represent distinct glial identities or different transcriptional states within these populations.”

(2) The authors present evidence for sex-specific neuronal and glia subtypes and find differential expression of specific yolk proteins and long non-coding RNAs. However, whether any of these differences are driven by other canonical sex-specific genes such as Fruitless (Fru) or Double-sex (Dbx) has not been reported or discussed. The authors must re-analyze their data for these genes and claim whether they have any contribution to sex-specific sub-clusters.

Thank you for pointing this out. We have made text changes and clarifications to highlight the expression of other canonical sex-specific genes. *Fru* was enriched in male nuclei as expected. Interestingly, *dbx* was enriched in female nuclei. It remains to be determined if these genes are mechanisms that may be driving sex-specific changes.

Lines 224-226: “Additionally, female nuclei were enriched for *dbx* (Supp Table 8). Male glial nuclei expressed higher levels of genes including the male-specific genes *lncRNA:rox1/2* and *fru* (Figure 5C; Supp Table 8) (Ryner et al., 1996; Amrein and Axel, 1997; Meller et al., 1997).”

Lines 237-239: “Male nuclei expressed higher levels of genes including the male-specific genes *lncRNA:rox1/2* and *fru* (Figure 5G; Supp Table 9) (Ryner et al., 1996; Amrein and Axel, 1997; Meller et al., 1997).”

Lines 428-431:” We found the expected differential expression of yolk proteins (*yp1, yp2, yp3*) enriched in female nuclei and the long non-coding RNAs *rox1/2* and *fru* enriched in male neuronal nuclei (Ryner et al., 1996; Amrein and Axel, 1997; Meller et al., 1997; Warren et al., 1979). Interestingly, we found *dbx* to be enriched in both glial and neuronal female nuclei.”

Lines 433-435: “It remains to be determined if these genes are driving these sex-specific differences in glia and neurons.”

(3) In Figure 6C, it is unclear whether the Ms-2A-LexA-expressing neurons of clusters 157 and 160 project to two different neuropils or share projects to both neuropils. However, it is not explicitly shown in the immunostaining data whether indeed there are two populations to begin with. The authors must check for cluster 157 and 160 specific markers (such as Dh44 and ple) and test whether they appear mutually exclusively in the Ms-2A-LexA-expressing neurons. The same reasoning would apply to the data shown in Figures 6D and 6E, where the authors must test whether the NPF and AstA expressing cells are indeed neurons from clusters 100 and 128, using orthogonal cluster markers to conclude that they are similar (or the same) neurons.

We changed the focus of the paragraph to confirm that these neurons indeed come from type II and that they target the central complex. Although due to the lack of reagents we cannot test the identity of each one of these neurons, we could make meaningful interpretations of the staining to validate our ideas about neuropeptidergic cells in the central complex. We made sure to mention the limitation of our experiment to avoid any wrong conclusions.

Minor points(1) Line 115 - "cluster that represents optic lobe neurons". How was this cluster identified?

We reexamined the most significant genes enriched in this cluster 124, and found they are *Rh2*, *ninaC*, *trpl*, and phototransduction related genes (Supplemental table 1). We reassigned the identity of this cluster as ocelli, which also express photoreceptor genes but can’t be easily removed during dissection. We modified the text as follows:

"We used known markers (Croset et al., 2018; Davie et al., 2018; Supp Table 2) to identify distinct cell types in the central brain, including glia, mushroom body neurons, olfactory projection neurons, clock neurons, Poxn+ neurons, serotonergic neurons, dopaminergic neurons, octopaminergic neurons, corazonergic neurons, hemocytes, and ocelli (Figure 1B, Supp. Table 1)."

(2) As the separation in Figure 1B is not obvious, annotated cell type clusters must be re-colored instead of being labelled as the exact dots are indistinguishable. This would especially be helpful for OCTY, SER, OPN, and CLK clusters.(3) Cluster labels in Figure 1C are barely visible and the font size must be increased for the reader. Recoloring the cluster identities and attaching a legend would again help in this case.

We recolored the atlas in Figure 1B, 1C and 1C’ and increased the font size in Figure 1C’.

(4) For Figure 4A, clusters should be labelled on the UMAP along with the legend as it is difficult for the reader to match identities using Seurat colors. The same is true for the UMAPs in Figure 5A.

Yes, we agree that labeling would improve readability and have done so for UMAPs in Figure 4A and 5A-A’’.

**Reviewer #2 (Recommendations for the authors):**
In this manuscript, Epiney et al., present a single-nucleus sequencing analysis of adult central brain neurons and glia Through the use of a ingenious permanent labeling technique, they are able to trace the progeny of T2 neuroblasts, which contribute significantly to the formation of the central complex. This transcriptomic dataset is the first of its kind and will likely serve as a valuable resource for future studies.The authors further explore this dataset through several analyses, including the characterization of neurotransmitter expression profiles in T2-derived neurons. However, the approach used to identify the identity of each neuron cluster could be more clearly articulated, and some of the authors' conclusions are more generalized - either already well-established or lacking sufficient support.Detailed comments:Abstract - "Our data support the hypothesis that each transcriptional cluster represents one or a few closely related neuron subtypes. - Is this a novel finding? If so, it would be helpful if the authors could explain why this is the case more clearly.

Our results are not generally novel, and many single cell/single nuclei RNA-seq papers have been published (more citations added to Introduction). Our work is novel in that we analyze Type 1 and Type 2 neuroblasts in the central brain.

Line 53 - In the introduction the authors should also reference other single-cell studies done in the *Drosophila* brain.

Done.

Line 59 - There are some typos here. The authors could also mention type zero.

Both done.

Figure 1 and Sup Table 1 - Authors show in sup table 1 the top cell markers by cluster but there is no correspondence between cluster number and identity. The authors do not say which known markers were used to give the identity to each cluster.

We have added the cell identity in the Supplemental Table 1. For the unknown cells, we left the column blank. We have also added a Supplemental Table 2 to show the markers we used to give identity to the clusters.

Supplementary Tables - For each table, more detailed information should be provided regarding what is being compared and the methods used for these comparisons.

We have added the methods we used in Seurat to generate each individual table.

Line 138 - Differential gene expression analysis between T1 and T2 glial progeny did not show differences across any glial cell types (Supp Table 4). - Was this comparison done per cluster? Is differential gene expression of top markers, which are anyway the genes that define each glial cell type, enough for this type of analysis?

Yes, we performed the differential expression analysis using all genes (i.e., not just marker defining) at a cluster-by-cluster resolution with results in Supplemental Table 4. We have edited the text to make this clarification.

Lines 139-141: “Differential gene expression analysis for all genes between T1 and T2 glial progeny did not show differences across any glial cell types or clusters (Supp Table 4).”

Line 146 - We identified T1-derived neurons by excluding cells co-expressing T2-specific. Markers FLP+/GFP+/RFP+ plus repo+ glial clusters. - Bioinformatically, correct?

Yes. We clarified the sentence as follows:

"We identified T1-derived neurons by bioinformatically excluding cells co-expressing T2-specific markers FLP+/GFP+/RFP+ plus repo+ glial clusters."

Line 156 - We found that each cluster strongly expressed a unique combination of genes. - As they are grouped by seurat in different clusters, why is this surprising?Line 175 - "top 10 significantly enriched genes gathered from each T2 neuron cluster" - can these lists be included?

Yes they are grouped by Seurat. We toned down the sentence and refer each combination of genes as cluster markers. We modified the sentences as follows:

Each unique combination of enriched genes could be referred to as cluster markers.

Line 211- How did the authors identify sex-biased clusters? How did the authors separate the samples/cells by sex? Was it done bioinformatically by the expression of certain genes? If so, which?

We collected male and female nuclei separately. We have added text in the methods section as follows:

"Equal amounts of male and female central brains (excluding optic lobes) were dissected at room temperature within 1 hour. The samples were flash-frozen in liquid nitrogen and stored separately at -80°.

In the first round, we pooled male and female brains together to select GFP+ nuclei and used particle-templated instant partitions to capture single nuclei to generate cDNA library (Fluent BioSciences, Waterton, MA). In the second and third rounds, RFP+ nuclei from male and female brains were collected separately. The split-pool method was then used to generate barcoded cDNA libraries from each individual nucleus."

Are there sex-specific differences in genes in glia other than genes that were previously known to be sex-specific?

We report the comprehensive list of sex-specific differences in gene expression for both glia and neurons in Supp tables 8 and 9.

Line 237 - When the authors mention "We conclude that male and female adult T2 neurons have sex-specific differences in gene expression within the same neuronal subtype" does this mean that these neurons are the same in male and in female brains, but they additionally specifically express sex-specific genes?

Yes, we report that male and females contain the same neurons defined by their transcriptional profile. It remains to be seen if this sex-specific differences changes how these same neuronal subtypes function between male and females. We have added additional text in the discussion to expand on this thought.

Lines 437-441: “It remains to be determined if these genes are driving sex-specific differences within glial and neuronal subtypes. These genes may reflect sex-specific differences in the adult central brain and may provide insight into how behavioral circuits are linked to sex-specific behaviors. Future work should aim to characterize and test these genes.”

Line 250 - The idea behind these sections "What is the relationship between neuropeptide expression and cluster identity?" "relation between cluster and morphology" lacks clarity. As clusters are defined based on principal component analysis, and the genes used to define a cluster are dependent on this method, there is no assumption that each cluster represents only one type of neuron or that it should include only neurons expressing the same neurotransmitter genes. Even if some clusters consist of a single neuron type, this should not be generalized to all clusters (and vice-versa).

Correct, we cannot determine from the transcriptome data whether distinct clusters will have different morphology. We have changed the focus of the question to address that we are confirming they come from type 2 and that they target the central complex while comparing to known cells that express the neuropeptide.

Line 265 - We first assayed the neuronal morphology of Ms+ neurons - why did the authors choose these neurons?

Resolved in main text: “we found that type II-derived Ms-2A-LexA-expressing neurons project to multiple layers of the dorsal fan-shaped body and the entire ellipsoid body, suggesting an unknown class of Ms+ neurons targeting to EB and/orFB".

Line 268 - "Currently we can't determine whether Ms+ neurons in clusters 157 and 160 project to different CX neuropils, or whether neurons from both clusters share projections into both neuropils. " - The purpose of this point is unclear.

Resolved in text: “we found that type II-derived Ms-2A-LexA-expressing neurons project to multiple layers of the dorsal fan-shaped body and the entire ellipsoid body, suggesting an unknown class of Ms+ neurons targeting to EB and/or FB”.

Line 279 - This analysis could be more explored.

Thank you for your feedback. As the comment was somewhat broad, we were unsure of the specific revisions needed and have therefore left the text unchanged.

Line 301 - The text regarding this section, and the description and details of respective figures should be proofread to ensure clarity.

Done.

Line 386 - Alternatively, co-expression may be due to background from RNAs released during dissociation. - RNA in soup could be bioinformatically analysed.

Correct. We opted to delete this sentence since our split-pool based method does not create background RNA expression. Additionally, the analysis is performed on scaled expression >2, and any background RNA is unlikely to yield such high expression.

Discussion - Some of the conclusions are a bit too general, suggesting that the results might be meaningful, but also acknowledging the possibility of artifacts. If the authors could refine this, it would strengthen the manuscript.

We are sorry but we are uncertain what you are asking; we don't know what you want us to refine. Our apologies for the misunderstanding.